# Underwater Leidenfrost nanochemistry for creation of size-tailored zinc peroxide cancer nanotherapeutics

Mady Elbahri[1,2,3], Ramzy Abdelaziz[1], Duygu Disci-Zayed[2,4], Shahin Homaeigohar[1], Justyna Sosna[5,6], Dieter Adam[5], Lorenz Kienle[7], Torben Dankwort[7] & Moheb Abdelaziz[1,2]

The dynamic underwater chemistry seen in nature is inspiring for the next generation of eco-friendly nanochemistry. In this context, green synthesis of size-tailored nanoparticles in a facile and scalable manner via a dynamic process is an interesting challenge. Simulating the volcano-induced dynamic chemistry of the deep ocean, here we demonstrate the Leidenfrost dynamic chemistry occurring in an underwater overheated confined zone as a new tool for customized creation of nanoclusters of zinc peroxide. The hydrodynamic nature of the phenomenon ensures eruption of the nanoclusters towards a much colder region, giving rise to growth of monodisperse, size-tailored nanoclusters. Such nanoparticles are investigated in terms of their cytotoxicity on suspension and adherent cells to prove their applicability as cancer nanotherapeutics. Our research can pave the way for employment of the dynamic green nanochemistry in facile, scalable fabrication of size-tailored nanoparticles for biomedical applications.

[1] Nanochemistry and Nanoengineering, School of Chemical Engineering, Department of Chemistry and Materials Science, Aalto University, Kemistintie 1, 00076 Aalto, Finland. [2] Nanochemistry and Nanoengineering, Institute for Materials Science, Faculty of Engineering, Kiel University, Kaiserstrasse 2, 24143 Kiel, Germany. [3] Center for Nanotechnology, Zewail City of Science and Technology, Sheikh Zayed District, 12588 Giza, Egypt. [4] Faculty of Technology and Bionics, Rhine-Waal University of Applied Sciences, Marie-Curie-Straße 1, 47533 Kleve, Germany. [5] Institute of Immunology, Kiel University, Michaelisstrasse 5, 24105 Kiel, Germany. [6] Department of Molecular Biology and Biochemistry, University of California, Irvine, California 92697, USA. [7] Synthesis and Real Structure, Institute for Materials Science, Faculty of Engineering, Kiel University, Kaiserstrasse 2, 24143 Kiel, Germany. Correspondence and requests for materials should be addressed to M.E. (email: mady.elbahri@aalto.fi).

In nature, dynamic chemistry is a unique discipline in terms of self-regulation and openness to flow conservation, enabling concurrent chemical synthesis and self-organization of minerals[1]. The fascinating minerals formed by flash deposition and eruption of supersaturated solutions on the bottom of the sea is an exemplary instance of the product of this versatile underwater dynamic chemistry[2]. Water is the main element of the dynamic processes in nature and an eco-friendly solvent. Thus, activation of water to act as an eco-friendly dynamic reactor for fabrication of size-tailored nanoparticles will open a new route for sustainable synthesis of functional materials. However, the particular challenge is to design a method able not only to sustainably synthesize nanoparticles[3–7] but also to tune their size for demanded functions in biomedicine, for instance in cancer treatment.

Nanomedicine[8–10] is widely explored for treatment of intricate diseases, yet comes with various challenges and questions. In this regard, exploration of the impact of nanoparticles (for example, those made of metal, metal oxide, and so on) on cell viability for therapeutic applications is going to be a highly developing research field. It is assumed here that peroxide nanoparticles (for example, zinc peroxide) could offer an effective nanotherapy. Peroxide has been introduced as a 'medical miracle'[11]. It acts as an oxygen supplier and thus is exploited in treatment of a wide variety of diseases induced by anaerobic and even cancerous cells. To employ peroxide nanoparticles in biomedical and therapeutic applications, it is preferred to synthesize them via a sustainable approach without involvement of additives and in a narrow size distribution. Fulfilment of these requirements is in fact the main challenge of using nanoparticles in biomedicine.

Here, we aim to mimic the natural recipe of dynamic chemistry to develop the new concept of underwater Leidenfrost dynamic chemistry. Within the frame of this new concept, employing the reactants in a confined space under hydrodynamic flow conditions, we are able to synthesize monodisperse $ZnO_2$ nanoparticles in an eco-friendly manner. The size of these nanoparticles can be optimally tailored and utilized for selective killing of cancerous cells.

## Results

**Underwater Leidenfrost phenomenon.** The Leidenfrost effect justifying the behaviour of a water droplet levitating on the top of a superheated surface at a temperature much higher than the boiling point of water is an established phenomenon. The Leidenfrost chemistry was introduced in 2007 (ref. 12). Since then, our research activities have notably contributed in resolving of the Leidenfrost phenomenon in terms of specification of the involved mechanism based on electrostatic and thermocapillary processes[13].

The Leidenfrost phenomenon encompasses several forms and is recognized by a minimum in the boiling curve, called as the Leidenfrost point[14]. With respect to each form, the phenomenon can be called differently. For instance, the Leidenfrost state of water droplet is known as the film boiling state in a pool boiling process. Thus, this reaction is not necessarily performed in a droplet and as patented by us the Leidenfrost chemistry can be carried out in other media, for example, in an aqueous solution[15]. In fact, the dynamic Leidenfrost phenomenon depends on the heat flux supplied by the interface and thickness of the thermal boundary layer of the fluid. Figure 1a and Supplementary Movie 1 visualize this effect where a small cold water film on a colder substrate is suddenly introduced to a superheated plate at 300 °C and the liquid is suddenly converted to a typical Leidenfrost droplet. On spending the incubation time, the adherent layer of

the liquid becomes superheated and hydrothermally explodes owing to its sudden conversion to steam. Meanwhile, primary vapour bubbles are formed and start to grow. The vaporization spreads out forward and the adherent liquid on the superheated substrate is expanded to vapour, causing a sudden drop in the hydrostatic pressure and the liquid is re-shaped as a droplet. The overheated state of the Leidenfrost phenomenon occurs and the droplet is now levitated. This phenomenon can take place on arbitrary surfaces, however if a large volume of water is present, though the phenomenon is ongoing under water, due to higher inertial forces, a droplet is not formed. To visualize this process, that is, the underwater Leidenfrost phenomenon, some loose carbon fibres were deposited on the bottom of a water-filled glass vessel. Then, the cold set-up was suddenly introduced to a hot plate, as shown by the snapshot images (Fig. 1b) and Supplementary Movie 2. It is obvious that the hydrothermal vents essential to release pressure at the interface form and the carbon fibres are splashed towards the colder region. The emergence and growth of the hydrodynamic vents are governed by the Rayleigh–Taylor instability owing to different densities of the overheated water (low density) and the cold water (high density) above. Following the generation of the overheated zone, a wave patterned vortex appears at the hot/cold fluids interface mainly due to the Kelvin–Helmholtz instability and thereby the black layer flows upward in the cold region. These movies and images (Supplementary Movie 2 and Fig. 1b, respectively) illustrate the hydrodynamic nature of the Leidenfrost phenomenon and the vapour generation underneath the water in an arbitrary form. Further evidences for the hydrodynamic nature of the Leidenfrost phenomenon will be presented in a subsequent section and discussed later.

**Formation of size-tailored nanoparticles.** Having thoroughly introduced the underwater Leidenfrost phenomenon wherein presence of an overheated zone at the interface is a vital pre-requisite, we hypothesize that mimicking the dynamic chemistry occurring deep in the ocean is feasible for lab-scaled synthesis and self-organization of nanoparticles. Separating the nucleation and growth steps, this approach could also be employed as a new strategy for generation of size-controlled nanoparticles. In more precise words, when nanochemistry occurs at the overheated zone, the formed nanoparticles assembled as nanoclusters erupt towards the much colder region for further growth. This tendency can be controlled and utilized for tailoring the size of the nano-particles, as schematically illustrated in Fig. 1c.

While the Leidenfrost charge-driven chemistry has been proven to be a versatile technique for synthesis of metal and metal oxide nanoparticles[12,13,15,16], we aim here to extend applicability of the technique to another class of materials, that is, metal peroxides such as $ZnO_2$. Conventional fabrication methods of $ZnO_2$ nanoparticles include hydrothermal synthesis, laser ablation, and sol–gel synthesis[17–19]. However, these methods involve chemicals such as different surfactants (for example, SDS, CTAB, OGM, and polyethylene glycol 200 (PEG 200)), thus could not be regarded purely green. In addition, as shown in TEM images, the generated particles are mostly aggregated, indicating nanoparticle instability[20]. To achieve ultrasmall and monodisperse $ZnO_2$ nanoparticles in the aqueous phase, recently, a new synthesis route was reported by Bergs *et al.*[21] In this approach, zinc acetate dihydrate was oxidized with hydrogen peroxide in an aqueous medium using high-pressure impinging-jet reactor. The high process pressure of 1,400 bar and a specially formed reaction chamber gave rise to short reaction times enabling fast nucleation and limited growth of nanoparticles, thus formation of ultrasmall $ZnO_2$

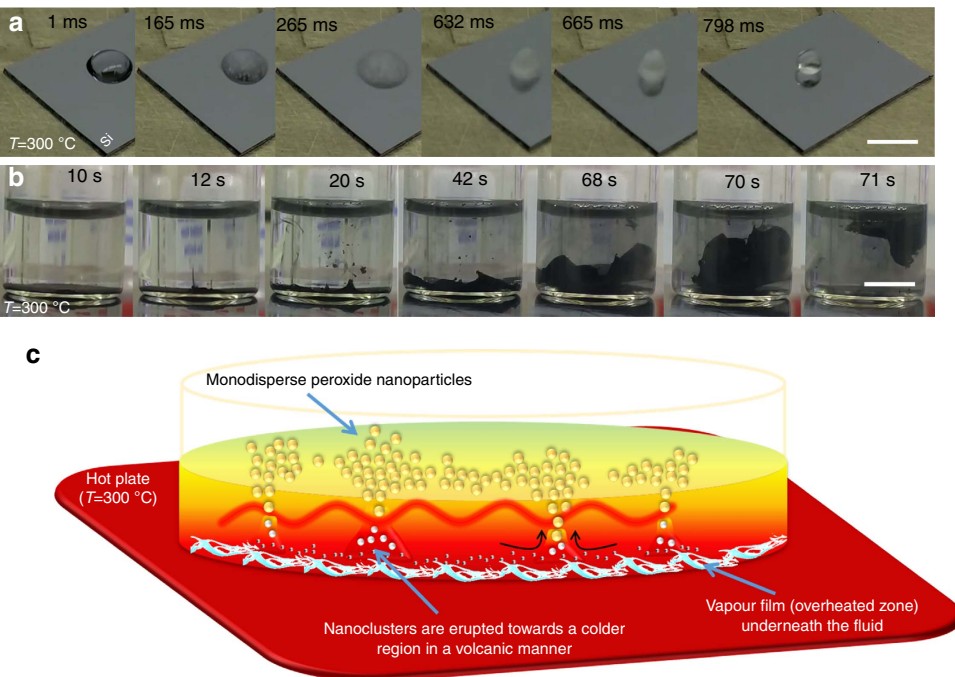

**Figure 1 | The underwater Leidenfrost phenomenon. (a,b)** The snapshots illustrate formation of a levitated droplet (70 μl) when a flattened cold droplet (film) mounted on a colder Si substrate is exposed to a superheated plate at 300 °C (**a**), and flight of carbon fibres (0.03 g) induced by the Leidenfrost effect under water (5 ml) and emergence of a vapour film flowing upward (**b**) (scale bars, 1 cm). (**c**) The schematic shows that when nanochemistry occurs at the overheated zone, the formed nanoparticles assembled as nanoclusters erupt towards the much colder region for further growth. This tendency can be controlled and utilized for tailoring the size of the nanoparticles.

nanoparticles. Despite the advantages of this technique in terms of formation of monodisperse nanoparticles, use of complicated and energy consuming instruments and chemicals such as the stabilizing agent of bis[2-(methacryloyloxy)ethyl]phosphate is challenging. Advantageous over such methods, here, we demonstrate a completely eco-friendly fabrication route for peroxide nanoparticles without involvement of chemicals for example, organic molecules and stabilizers. More importantly, the entire process is accomplished in a small reactor, that is, a water bath exposed to a hot plate, implying simplicity and cost/energy efficiency of the process.

Thus, through this study, applicability of the Leidenfrost dynamic chemistry in synthesis of peroxide monodisperse nanoparticles is proved. Moreover, the underwater dynamic nature of the process is to be demonstrated and visualized. To do so, 9 ml of a zinc acetate aqueous solution (10 mM; hereafter this experiment will be referred within the manuscript as the 10 mM experiment) mixed with 1 ml hydrogen peroxide in a Petri dish was suddenly introduced to a superheated plate with a temperature of 300 °C. Figure 2a (extracted from Supplementary Movie 3) demonstrates snapshots of the Leidenfrost dynamic chemistry in a Petri dish (the 10 mM experiment, but at a larger volume of 50 ml). In this figure, eruption of the clusters from the overheated region to the cold one after an incubation time of 40 s is clearly seen. The naked eye observation of the formed clusters (Supplementary Movies 3 and 4) implies also a colour transformation of the solution from colourless to milky (Fig. 2a). SEM images of the fished clusters from the cold interface (Fig. 2b) demonstrate the quasi monodisperse nanoparticles arranged as self-assembled groups with a narrow size distribution of 426 nm, as measured by a particle size analyser (Fig. 2c). X-ray diffraction (XRD) patterns (Fig. 2d) of the synthesized nanoparticles reveal two distinct reflections at $2\theta = 31.79°$ and $36.87°$. The absence of

the ZnO characteristic reflection at $34.422°$ implies the solely formation of $ZnO_2$ nanoparticles[22–24]. Accordingly, the XRD pattern could be completely assigned to $ZnO_2$.

Scaling up the system while tailoring the nanoparticles size was experimentally realized in a 100 ml beaker containing 50 ml zinc acetate (50 mM; hereafter this experiment will be referred within the manuscript as the 50 mM experiment) aqueous solutions per 5 ml hydrogen peroxide, (Fig. 2e, Supplementary Movie 5). It is noteworthy that the water used in all experiments as the main solvent was double de-ionized water (conductivity of $0.055\,\mu S\,cm^{-1}$). As shown in Fig. 2f (and Supplementary Fig. 1), the experiment demonstrated the possibility of facile large-scale production of size-tailored nanoparticles, with a narrow size distribution around 126 nm. As seen in Fig. 2f, the as-synthesized nanoparticles are very uniform. This approach allows fabrication of a collection of size-tailored self-assembled nanoclusters, as illustrated in Fig. 2g–i. For instance, using 5, 20, and 70 mM zinc acetate aqueous solutions, the average particle size is tuned as 680, 220, and 70 nm, respectively (Supplementary Figs 2 and 3). The high monodispersity of the synthesized nanoparticles is an important advantage for the presented version of the Leidenfrost process over the classic one relying on a superfast spinning and wandering droplet. The complicated dynamics of a Leidenfrost droplet hampers creation of monodisperse particles (Supplementary Fig. 4). It is noteworthy that the vessel procedure enables us to scale up the technique while tracking the underwater chemistry in a more precise manner.

The driving force of the Leidenfrost dynamic chemistry is supplied via a combination of charge generation in an overheated confined zone and a thermocapillary process. Figure 3a demonstrates charge separation in the aqueous solution (the 10 mM experiment; 50 ml) subject to a fast heating process. As seen in this figure, negatively charged water is created as similarly

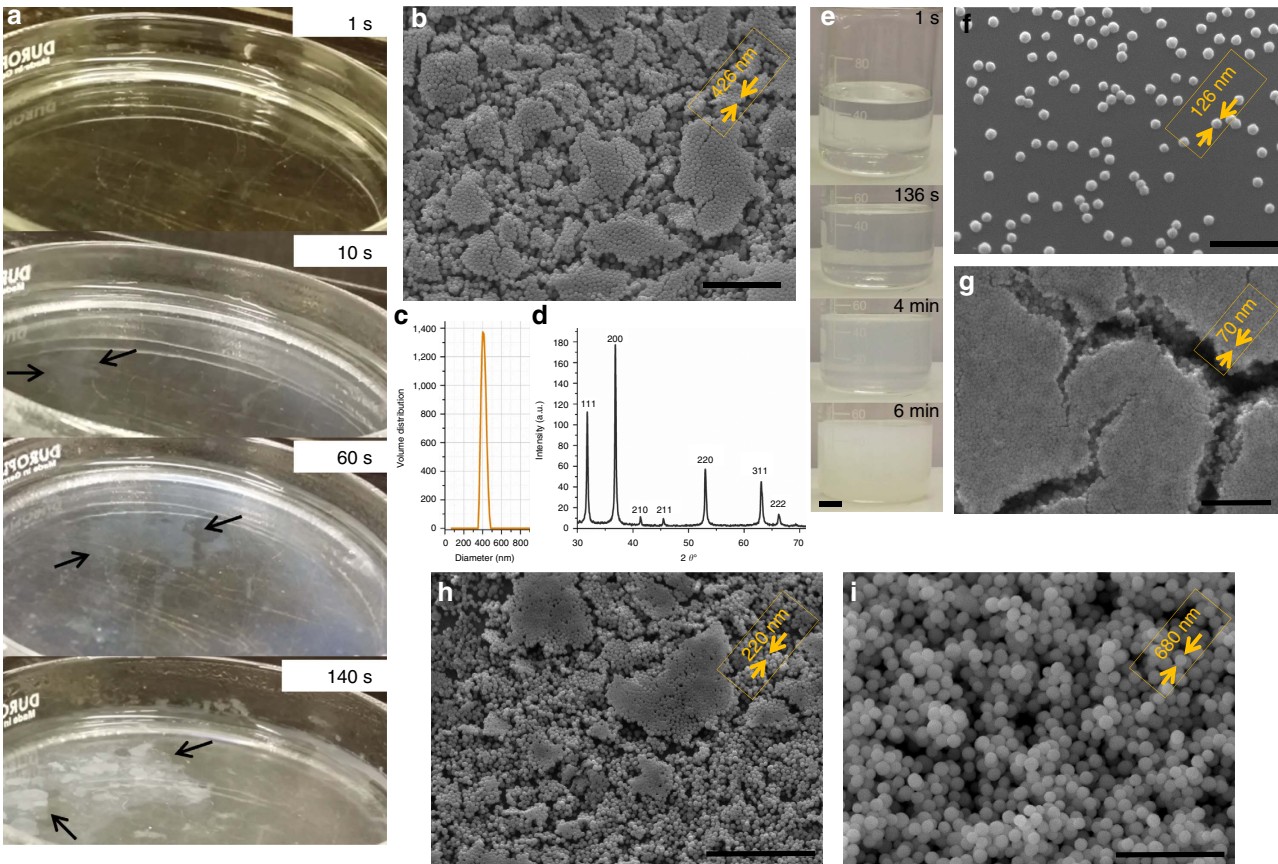

**Figure 2 | Fabrication of monodisperse ZnO₂ nanoparticles via the Leidenfrost charge-driven chemistry.** (**a**) The snapshots visualize the Leidenfrost chemistry in a Petri dish and eruption of the formed clusters, specified by the black arrows (extracted from Supplementary Movie 3 based on the 10 mM experiment but at a larger volume of 50 ml); (**b**) SEM image depicting morphology of the synthesized monodisperse ZnO₂ nanoparticles obtained from the 10 mM experiment (scale bar, 5 μm); (**c**) Particle size distribution of the ZnO₂ nanoparticles; (**d**) XRD pattern of the synthesized ZnO₂ nanoparticles; (**e**) The snapshots visualize the Leidenfrost chemistry in a beaker (extracted from Supplementary Movie 5) (scale bar, 1 cm); SEM images of the synthesized monodisperse ZnO₂ nanoparticles obtained from the 50 mM experiment: size-tailored self-assembled monodisperse nanoclusters in different average particle sizes of (**f**) 126 nm, (**g**) ∼70 nm (scale bars for **f,g** are 1 μm), (**h**) ∼220 nm, (**i**) ∼680 nm (depending on the precursor zinc acetate solution concentration) (scale bars 5 μm (**h,i**)).

observed in our previous experiment based on the Leidenfrost droplet[13]. The details of the charge measurements have been described elsewhere[13], but in general, a metal tungsten tip was connected to a grounded electrometer and then placed in an aqueous solution. The tip was located at a distance of 1 mm from the bottom of the aqueous solution containing Petri dish exposed to a high temperature of 300 °C and charges were registered. The charge generation is governed by detachment of the water surface layer adjacent to the superheated surface and rupture of the double electric layer on the surface. This mechanism was proved for the first time by Volta and George[25], and further studied by Faraday[26] and Lenard[27], and has been recently demonstrated by us to occur at the Leidenfrost temperature[13]. This effect is diminished under a conventional heating process, as demonstrated in Fig. 3a. Such a discrepancy between the effect of different modes of heating supports an earlier finding on importance of the fast heating process in charge generation and its relevant nanochemistry[13]. To highlight the importance of this point while characterizing different boiling regimes through the charge concept, we performed additional experiments, whereby the temperature of the hot plate thus the water bath was correlated to the induced charge in the bath reactor (Supplementary Fig. 5). Accordingly, we could register the temperatures relevant to the transition and onset of the Leidenfrost point at 150 and 250 °C, respectively. The fast evaporation event at this stage is characterized by creation of the negatively charged fluid as shown in Supplementary Fig. 5. When the temperature rises over 150 °C, negative charges form in the medium. It is noteworthy that no charges were registered for de-ionized water at room temperature or even at boiling temperature. As much as temperature increases, that is, different regimes of boiling, the magnitude of charges rises and thereby conversion of the salt precursor to the peroxide particles is promoted, as shown in the SEM images of Supplementary Fig. 5. These results in fact highlight the importance of the Leidenfrost state and its relevance to the nanochemistry.

In addition to the notable role of the charge-driven chemistry, the hydrodynamic nature of the underwater Leidenfrost process must be clarified. The confined dynamic chemistry underneath the water is self-regulated by a thermocapillary driven hydrodynamic response, as proven by snapshots of the IR camera. In particular, Fig. 3b shows the development of the confined state and temperature gradient within a Petri dish that has been suddenly introduced to a hot plate at 300 °C. It is obvious that a temperature gradient in analogous to that seen in the Leidenfrost droplet[13] has been established. The highest temperature

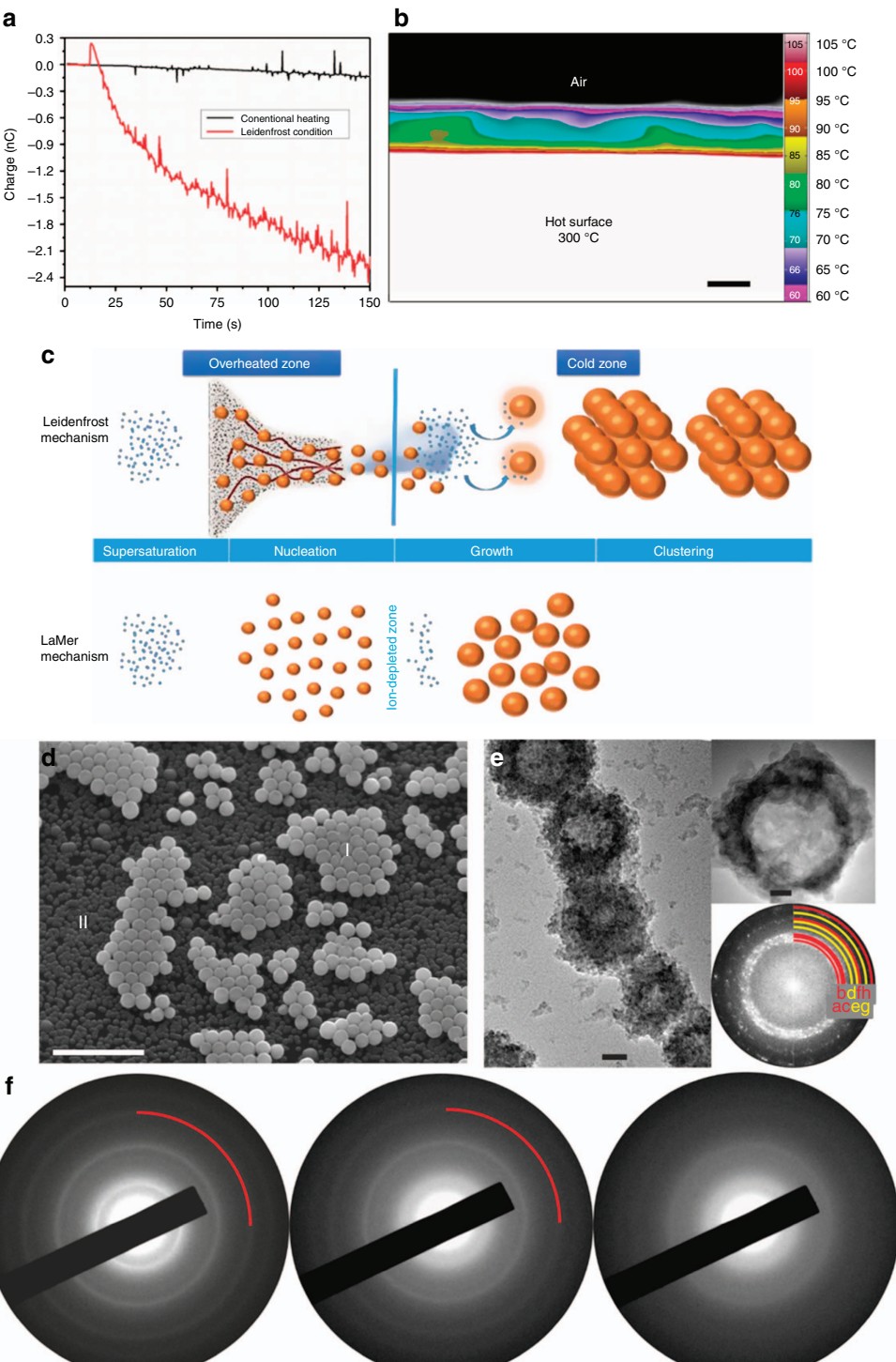

**Figure 3 | Charge separation and hydrodynamic instability in the Petri dish at the Leidenfrost condition.** (**a**) Charge curves of the aqueous solution (the 10 mM experiment, 50 ml) measured inside the Petri dish at conventional heating (heating from 25 up to 300 °C) compared to that measured in the Petri dish at 300 °C (that is, the Leidenfrost temperature); (**b**) Cross-sectional thermographic view of the solution inside the Petri dish at 300 °C shows different colours corresponding to different temperatures (in °C) (scale bar, 3 mm); (**c**) A schematic illustrating the main stages of the Leidenfrost mechanism in our study compared to those of the La Mer mechanism. In our technique, very high concentration of ions in the overheated zone and very fast nucleation lead to a narrow size distribution and large amount of the fabricated particles. On the other hand, huge availability of reactants within the cold region, and reduction of ions that is catalysed by the already formed nanoclusters result in the uniform growth of the particles whose size is tailored by the solute concentration. (**d**) SEM image of the dual sized $ZnO_2$ nanoparticles (marked by I and II signs, respectively) synthesized during the eruption and growth process (scale bar, 2 μm); (**e**) TEM micrographs depicting a representative overview of the sample (left; scale bar, 20 nm) and HRTEM micrograph depicting ca. 50 nm hollow spheres produced by electron beam impact (upper right; scale bar, 10 nm), and corresponding Fourier Transform (lower right) signifying clustering of nanoparticles (labels a through f correspond to weak and strong reflections of ZnO and $ZnO_2$ crystalline planes); (**f**) Time-resolved ED data under low-dose illumination (∼1s illumination time for each image; the marks are attributed to a reflection at $d = 1.756$ Å being characteristic for $ZnO_2$).

                                                                           

exceeding 100 °C exists in the overheated zone of the two-phase region (liquid/steam). Also, induced by the temperature gradient, a wavy interface between the lower (hot) and upper (less hot or 'colder') part of the fluid is created, which is responsible for the hydrodynamic response and eruption of the clusters (as visualized in Fig. 2a and Supplementary Movie 3). In this context, the gradient environments act as a generalized force to separate the nucleation and macroscopic growth of the clusters. Under this circumstance, the overheated zone provides an optimal condition for a confined reaction wherein the high evaporation rate leads to local supersaturation hence nucleation and primary growth of the clusters. Induced by the hydrodynamic response, the primary formed clusters, acting as seeds, will then erupt to the colder region wherein further interaction and growth happen. Separating nucleation and growth steps, this scenario, which is a kinetically modified version of the well-known La Mer mechanism[28,29], would give rise to a new type of dynamic chemistry for creation of nanocrystals with a controlled narrow size distribution. In the La Mer mechanism, a short burst of nucleation after saturation concentration leads to fast formation of a large number of clusters, which then grow in a slow rate in an ion-depleted zone under steady state conditions. In this context, there is no physical barrier separating the nucleation and growth steps. In contrary, the nucleation and growth under the Leidenfrost condition are totally separated and occur in two different zones. While nucleation and primary growth occur in the overheated zone, by eruption to the colder region the clusters grow in a controlled manner, that is, nucleation and growth take place in completely isolated regions. Thus, our proposed system is indeed a non-steady one. Here we refer to the corresponding differential equation of the mass balance equation for the number of the formed stable nuclei, $n_+$, applicable to the La Mer model (equation 1)[30]:

$$v_+ \frac{\mathrm{d}n_+}{\mathrm{d}t} + \dot{v}_+ n_+ - QV_m = 0, \tag{1}$$

where $v_+$ is the minimum particle volume of the stable nuclei, $\dot{v}_+$ is the mean volume growth rate of the stable nuclei, $Q$ is the supply rate of solute in mol per unit time and $V_m$ is the molar volume of the solid. In our non-steady Leidenfrost model, since $\mathrm{d}n_+/\mathrm{d}t$ as well as $V_m$ are irrespective of m, there is no competition between nucleation and growth of the particles. In our technique, it is speculated that very high concentration of ions in the overheated zone and very fast nucleation are responsible for the narrow size distribution and large quantity of the fabricated particles. In this regard, huge availability of reactants within the cold region and reduction of ions that is facilitated and catalysed by the already formed nanoclusters lead to uniform growth of the particles whose size is tailored by the solute concentration, as previously shown in Fig. 2g–i. The schematic illustrated in Fig. 3c, depicts the fundamental differences between the La Mer mechanism and ours.

This hypothesis was confirmed by fishing the erupted nanoclusters of the 10 mM experiment on a silicon substrate and a TEM grid at different time intervals. SEM image of the clusters (Fig. 3d) reveals presence of tiny and large particles formed at onset of the experiment (after 10 s). The time-dependence growth is further confirmed by TEM observation of ultrafine nucleated zinc peroxide clusters, formed during the 10 mM experiment (after 1 s). With respect to the chemical composition of $ZnO_2$, it is worth mentioning that the products were unstable against the irradiating beam, quickly transforming from compact nanoparticles to hollow spheres of a $ZnO/ZnO_2$ mixture after irradiation. In Fig. 3e, the final state of the particles is depicted while images of intermediate stages can be found in

Supplementary Fig. 6. High-resolution TEM (HRTEM) and fast fourier transform (FFT) images of a representative ca. 50 nm hollow nanoparticle further witnesses that the spheres consist of randomly oriented clusters. In absence of reflections being characteristic for one component, most reflections can be indexed assuming ZnO and $ZnO_2$, respectively. However, $d$-values of 2.6027 Å and 1.9107 Å ($(002)_{ZnO}$ and $(012)_{ZnO}$, respectively) are only consistent with ZnO structure, and $d = 1.722$, 1.9886, and 2.1784 Å ($(022)_{ZnO_2}$, $(112)_{ZnO_2}$ and $(102)_{ZnO_2}$) are characteristic for $ZnO_2$. From the FFT, the strong peaks with $d$-spacing of 2.65 and 1.93 Å (marked b and f in Fig. 3e bottom right) imply the presence of ZnO nanoparticles as the major component. Whereas, the weak peaks with $d$-spacing of 1.77, 2.01, and 2.19 Å (marked yellow d, e, and g in Fig. 3e right, bottom) are attributed to the minor quantity of $ZnO_2$.

As the above-described products transform by electron beam irradiation, the pristine state was investigated using electron diffraction at a low dose and short exposure time. Time-resolved series of electron diffraction (ED) patterns are shown in Fig. 3f. As seen in the figure, clearly sharp rings are found at the start of the measurements, including a reflection at $d = 1.756$ Å being characteristic for $ZnO_2$ (see red mark). However, after few seconds the sample reacts and crystallinity decreases rapidly. Based on the XRD and TEM results, the observed particles are $ZnO_2$ particles initially, which quickly transform to hollow spheres made of ZnO and $ZnO_2$ nanoparticles on electron beam irradiation.

**$ZnO_2$ particles as nanotherapeutics**. Having synthesized the monodisperse $ZnO_2$ particles, we explored their potential in treatment of intricate diseases. Nanoparticles have been already suggested as a tool for cancer therapy[31–33]. The relevant studies have demonstrated that the cytotoxic effect of nanoparticles against cancer cells depends strongly not only on the size but also on the metabolic activities of the cells[34–37]. Accordingly, here for the first time, we performed a series of initial experiments to determine the impact of $ZnO_2$ nanoparticles on the survival of cancer, and also normal, healthy cells. In particular, we investigated the impact of $ZnO_2$ nanoparticles of different sizes on the survival of freshly isolated, healthy peripheral blood mononuclear cells (PBMCs) in comparison to cancer suspension cells such as leukaemic Jurkat T cells, U937 lymphoma cells, as well on adherent tumour cell lines such as HT29 (human colorectal carcinoma), Panc89 (human pancreatic adenocarcinoma), and L929Ts (murine fibrosarcoma). In the initial experiments, we measured loss of membrane integrity as the indicator of cell death, determined by uptake of membrane impermeable dye—propidium iodide. Size- and concentration-dependent cytotoxicity profiles for the above-mentioned cell lines and for PBMC were determined. The obtained results for the suspension cells and adherent cell lines are shown in Fig. 4a–f, respectively. As seen in the figures, cytotoxicity of the $ZnO_2$ nanoparticles depends mainly on the particle size and the target cells' type as adherent or suspension. With PBMC and suspension cells such as Jurkat and U937, the 126-nm-sized nanoparticles created a more pronounced toxic effect and caused more cell death than the particles of 426 nm in size did (Fig. 4a–c). However, in the case of adherent cells such as Panc89 and HT29, the smaller nanoparticles showed a higher toxic effect only at concentrations above 200 μg ml$^{-1}$. At concentrations equal or below 200 μg ml$^{-1}$, the 426 nm particles were more effective. In the case of murine L929Ts, regardless of the nanoparticles concentration, the larger particles were more lethal to the cells. Strikingly, healthy PBMCs were less affected by cytotoxic action

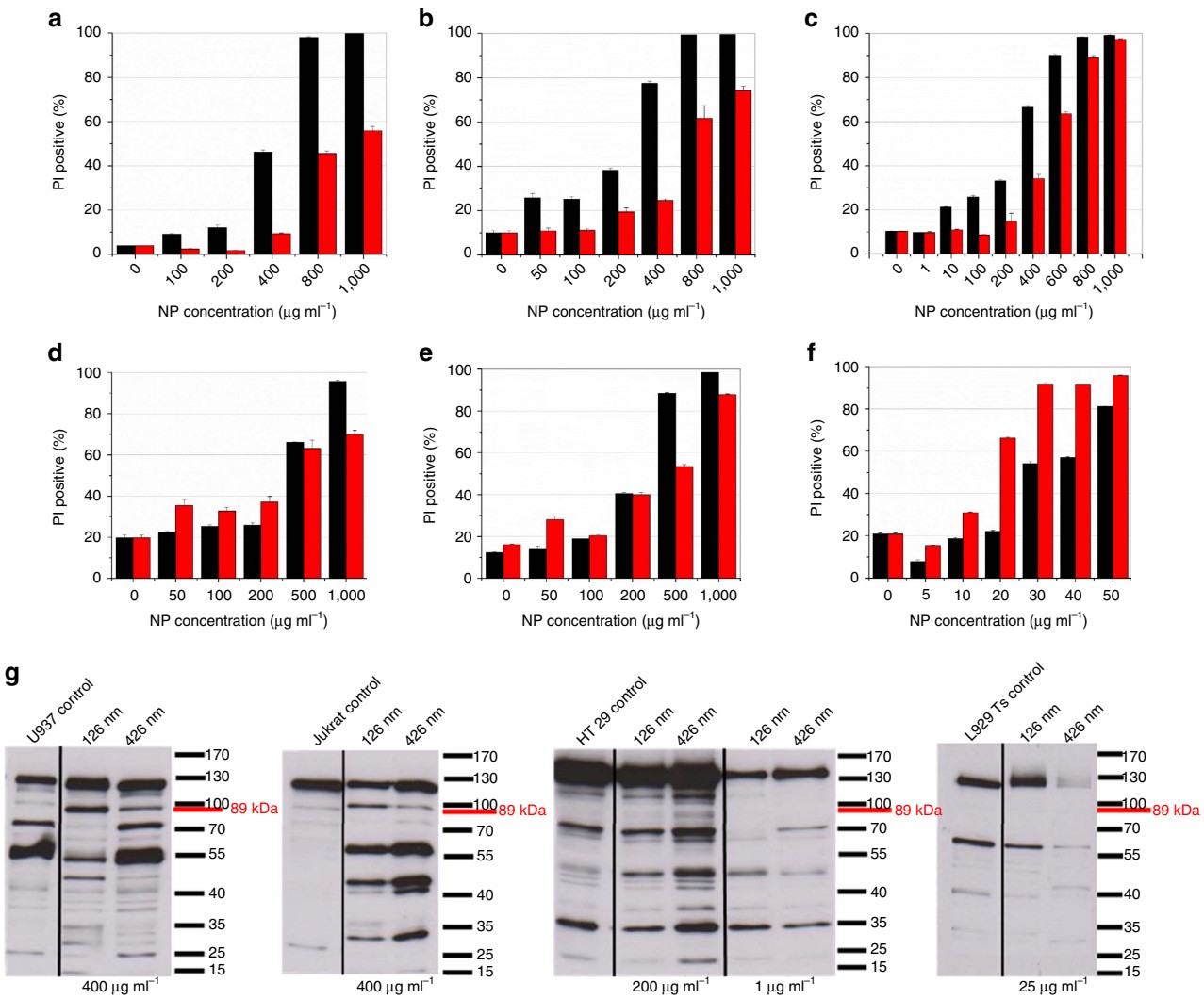

**Figure 4 | Cell mortality induced by cytotoxic effect of ZnO₂ nanoparticles.** Flow cytometric analysis of cells (**a**) Jurkat ATCC, (**b**) PBMC, (**c**) U937, (**d**) HT29, (**e**) Panc89, (**f**) L929Ts stained with PI (PI-positive fraction) treated with the 126 and 426 nm ZnO₂ nanoparticles (black and red columns, respectively) at various concentrations for 24 h, '0' concentration represents blank (control) samples containing the cell culture media but in absence of ZnO₂ nanoparticles. (**g**) Western blot analysis of PARP-1, representing apoptotic and necrotic events for U937, Jurkat ATCC, HT29, and L929Ts cell lines. Note: In this Figure, one representative experiment out of three with three parallel determinations each is shown. In addition, error bars, calculated using the STD function of Microsoft Excel, indicate the corresponding s.d.'s.

of the 426 nm ZnO₂ nanoparticles than U937 or L929Ts cells. However, for Jurkat, HT29, and Panc89 cells, the cytotoxic response induced by both sizes of nanoparticles was comparable to that for PBMC (Fig. 4a,d,e versus 4b).

To gain further insight into the cell death mechanisms (that is, whether cell death occurred by classical apoptosis or by other forms of regulated cell death such as regulated necrosis[38]), we investigated the cleavage of poly(ADP-ribose) polymerase-1 (PARP-1) in U937, Jurkat, HT29, and L929Ts cells treated with ZnO₂ nanoparticles (Fig. 4g). In apoptotic cells, PARP-1 is inactivated by caspase-3-dependent cleavage of the full-length 116-kDa protein to an 89-kDa product. However, in the cells undergoing necroptosis (for example, TNF-treated L929Ts cells), PARP-1 displays an atypical size shift/disappearance of the mature, uncleaved protein[39]. As shown in Fig. 4g, all the cell lines except L929Ts displayed an increase of the 89-kDa band, indicating that treatment with ZnO₂ nanoparticles induced apoptosis.

In contrast, in the ZnO₂ nanoparticle-treated L929Ts cells, the 89-kDa band was absent but rather a shift/disappearance of

the full-length PARP-1 protein was detectable (Fig. 4g). This indicated that ZnO₂ nanoparticles have the potential to induce both apoptotic and necroptotic cell death. This is particularly relevant as the induction of necroptosis in cancer cells represents a novel and only marginally explored option for future cancer therapies[40–42].

These experiments indicate that ZnO₂ nanoparticles have the potential to kill tumour cells by apoptotic and non-apoptotic mechanisms. At present, the efficacy of killing seems to be dependent on the particle size and on the type of the examined tumour cell. It is noteworthy that healthy PBMCs were also susceptible to negligible cytotoxic effects of the 426 nm ZnO₂ nanoparticles at 50 μg, compared to certain tumour cells such as L929Ts.

**Discussion**
All in all, we introduced the Leidenfrost charge-driven chemistry that could occur on an arbitrary overheated confined zone as a new artificial dynamic chemistry discipline. Mimicking the dynamic

chemistry in nature, more specifically near volcano gates of the deep ocean, the approach enables us to fabricate peroxide particles in controlled and tailored size. It is noteworthy that in a broader spectrum, our green nanofabrication method has also shown a promising applicability in synthesis of a diverse range of other functional nanomaterials including metals and metal oxides. Our results clearly show that our sustainably synthesized, size-controlled nanoparticles can play a key role in the next generation of cancer nanotherapeutics. In this report on the cytotoxic effect of zinc peroxide particles, we demonstrate that such an adverse effect on human cells, cancer suspension cells, and adherent tumour cells depends strongly on the size of the particles and the cell physiology, as previously reported for oxide particles. Regarding the former dependency, even though the origin of the cytotoxicity caused by the nanoparticles is not completely understood, generation of reactive oxygen species is thought to be the cause. Size and shape of the nanoparticles would strongly influence their toxic potential. In general, as the particle size decreases, more reactive oxygen species generation is expected due to increased surface defects of nanoparticles, decreased nanocrystal quality, and higher electron donor–acceptor impurities. However, as aforementioned, cytotoxicity is not only linked to nanoparticle characteristics, but also depends on the cell type. This fact was the case in our research and we observed different cytotoxic profiles for suspension and adherent cells for different sizes of $ZnO_2$ nanoparticles. Considering uniqueness of our work in terms of the type of the nanoparticles being used as cancer therapeutics, that is, peroxides, it is difficult to clarify this effect with no previous knowledge concerning similar systems.

In contrast to $ZnO_2$, there are numerous studies based on cytotoxic effect of ZnO that could be regarded for the sake of comparison. In one of the most similar studies, Lin et al.[43] employed toxic effect of ZnO nanoparticles (70 and 420 nm) on human bronchoalveolar carcinoma-derived cells (A549). They found that with no significant difference, the particles of either size can reduce the cell viability in a dose and time dependent manner (8–18 $\mu g\, ml^{-1}$) by inducing oxidative stress. In another related study, Kundu et al.[44] studied the cytotoxic effects of ZnO nanoparticles (100–120 nm) on HT29 colon carcinoma cells and PBMCs. They reported that ZnO nanoparticles preferentially kill HT29 tumour cells over "normal" PBMCs. This result is in contrast to our findings as $ZnO_2$ nanoparticles, regardless of size, kill HT29 (as well as for Jurkat and Panc89) cells, as much as they kill PBMCs. Such a discrepancy is likely due to different interaction mode of $ZnO_2$ and ZnO nanoparticles with cells. Considering the absence of any relevant study about biological performance of $ZnO_2$ nanoparticles, we are unable to further discuss this point at the current moment. However, our research is ongoing, and we hope to discover the probable mechanism involved in the cytotoxic effect of $ZnO_2$ particles to various cell types.

## Methods
**Sample synthesis.** For the sake of synthesis of monodisperse zinc peroxide nanoparticles through the Leidenfrost dynamic chemistry, zinc acetate (Sigma Aldrich, USA) aqueous solutions mixed with hydrogen peroxide (Sigma Aldrich, USA) in a Petri dish or beaker were suddenly introduced to a superheated plate with a temperature of 300 °C. The details in terms of concentration and volume of the reactants were previously mentioned in the Results section.

**Particle size distribution measurements.** The size distribution of $ZnO_2$ nanoparticles was determined using a ZetaSizer Particle Size Analyzer (Nano-ZS equipped with a red laser (633 nm, 4 mW) and a detection angle of 173°, Malvern Instrument Ltd, Malvern, England) and an avalanche photodiode detector. The size of the nanoparticles in water suspensions at ambient temperature was measured based on the dynamic light scattering method. The suspensions were the Leidenfrost aqueous suspensions (2 ml; 1:1 diluted with de-ionized water) made after 40 s

exposure of the precursor solution (5, 10, 20, and 70 mM inversely proportional to the particle size) to a 300 °C hot plate. For each sample suspension, dynamic light scattering measurements were carried out in Plastibrand semi-micro poly(methyl methacrylate) cuvettes at 25 °C with a fixed run time of 20 s. The scattering angle was set at 90°. The instrument recorded the intensity autocorrelation function, which was transformed into volume functions to obtain size information. The autocorrelation curves were fitted by the Malvern software (Malvern DTS 5.10 software). Two methods of analysis were used, including cumulants analysis to determine a mean size and polydispersity index and distribution analysis to determine actual size distribution. Using the fitted correlation functions, diffusion coefficients were obtained that were associated to hydrodynamic diameter via the Stokes–Einstein equation.

Moreover, histograms of average particle size were determined from 100 random particles in an arbitrarily chosen area in the SEM images (Fig. 2b,f–i).

**TEM and SEM characterization.** TEM measurements were carried out using a FEI Tecnai F30 G2 STwin (FEG cathode, 300 kV, $C_s = 1.2$ mm) equipped with a Si/Li EDX detector (EDAX-system). Further, a JEOL JEM-2100 (200 kV, $LaB_6$, $C_s = 1.0$ mm) was used for TEM observations at a lower dose. Particles were transferred to a Cu lacey carbon TEM grid. SEM measurements were carried out using a Phillips XL 30 and a Field Emission SEM ZEISS ULTRA plus equipped with Oxford Instruments INCA-X-act EDS.

**XRD and infrared imaging investigation.** XRD of the particles were obtained using a Seifert XRD 3,000 PTS (RICH. SEIFERT & Co GmbH) with a two-circuit goniometer. All measurements were carried out with Cu-K$\alpha$ radiation ($\lambda = 1.5418$ Å) operating at 40 kV and 30 mA at ambient temperature. To ascertain that monochromatic ray radiation is used in the measurements, a Ni filter or a monochromator was employed. The measurements were performed over an angular range of 25–80° to both planar thin films deposited on a glass substrate. The software used in the analysis and background correction was Rayflex. InfraTec PIR uc 180 with a rate of 24 Hz and a geometric resolution of 160 × 120 IR pixels was used for thermographic analysis in the 10 mM experiment.

**Biological tests.** HT29, Jurkat, and U937 cell lines were purchased from ATCC (catalogue numbers: HT29: HTB-38, Jurkat: TIB-152, U937: CRL-1593.2). L929Ts is a tumour necrosis factor-related apoptosis-inducing ligand-sensitive L929 sub-line derived in the laboratory of D.A.[45]. The cell line Panc89 has been described previously and was kindly provided by H. Kalthoff, Kiel[46]. PBMCs were isolated by Ficoll density gradient centrifugation from leukocyte concentrates. Leukocyte concentrates of healthy blood donors were obtained from the Institute for Transfusion Medicine of the University Hospital Schleswig–Holstein. All cell lines were frozen in aliquots at early passages, and for each experiment, cells were rethawed from those early aliquots, tested for being free of mycoplasma contamination, used for experiments for a maximum of 4 weeks and then discarded.

PBMCs, Jurkat, and U937 cells were cultured in RPMI 1640 (Thermo Fisher Scientific, Dreieich, Germany) with 10% fetal bovine serum, 10 mM Hepes, 10 IU $ml^{-1}$ penicillin + streptomycin, respectively, and incubated at 37 °C under a humidified atmosphere with 5% $CO_2$. As soon as they reached the required confluence, they were harvested and seeded to 12-well plates in 1 ml volume with $1 \times 10^6$ cells per ml density and stimulated by different concentrations of nanoparticles. Other cell lines were cultured in DMEM (Thermo Fisher Scientific) with 5 IU $ml^{-1}$ penicillin streptomycin/L-glutamine (Thermo Fisher Scientific), 10% fetal bovine serum and incubated at 37 °C under a humidified atmosphere with 5% $CO_2$. On reaching the required confluence, they were harvested and seeded to 12-well plates in 1 ml volume with $15 \times 10^4$ cells per ml density. The density has been optimized for all cell lines due to their confluence size in well plates.

The synthesized nanoparticles were washed three times with de-ionized water and sedimented by centrifugation. Centrifugation of the nanoparticles was performed on a Hettich EBA 20 centrifuge at 3,461 g for 90 min. Subsequently, the nanoparticles were washed through sonication for 45 min. The sedimented nanoparticles were then dried at 40 °C in ceramic crucibles to remove the excess water. The dried $ZnO_2$ nanoparticles were weighted and re-suspended in sterilized water to make the desired stock solution concentration. At last, the nanoparticles were introduced into cell cultures in a total volume of 1 ml. The stimulated cell cultures were incubated in a humidified incubator at 37 °C and 5% w/v $CO_2$ for 24 h to conduct further characterizations.

For all cell lines, uptake of propidium iodide (PI) measured by flow cytometry was used to determine the fraction of dead cells. PI stains the dying cells when they lose their membrane integrity at late apoptotic and/or necrotic stage[47]. After 24 h of stimulation, cells were washed with ice-cold PBS and centrifuged at 400 g, 4 °C for 5 min. For adherent cells, an additional detachment procedure with accutase was applied. The cells were re-suspended in PBS per 5 mM EDTA containing 2 $\mu g\, ml^{-1}$ of PI, and the red fluorescence was measured on a FACSCalibur flow cytometer (Becton Dickinson, Heidelberg, Germany). Each measurement was done in triplets with 10,000 gated events. For western blots, cells were harvested after treatment and lysed at 4 °C in TNE buffer (50 mM Tris pH 8.0, 1% v/v NP40,

2 mM EDTA) supplemented with 10 µg ml$^{-1}$ pepstatin/aprotinin/leupeptin, 1 mM sodium orthovanadate, and 5 mM NaF. Identical amounts of protein per lane were resolved by electrophoresis on SDS–polyacrylamide gels. After electrophoretic transfer to nitrocellulose, reactive proteins were detected using an antibody specific for PARP-1 (9542, Cell Signaling, Danvers, MA, USA, 1:1,000 dilution), horseradish peroxidase-coupled AffiniPure Goat Anti-Rabbit IgG (H + L) secondary antibodies (111-035-003, Jackson Immuno Research Laboratories, West Grove, PA, USA, 1:10,000 dilution), and the ECL detection kit (GE Healthcare, Munich, Germany). Equal loading as well as efficiency of transfer was verified by Ponceau S staining. All uncropped immunoblots are shown in Supplementary Fig. 7.

**Data availability.** The authors declare that the data supporting the findings of this study are available within this paper and its Supplementary Information file, or from the corresponding author.

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

## Acknowledgements

M.E. thanks the School of Chemical Engineering at Aalto University, Finland, and the Initiative and Networking Fund of the Helmholtz Association (grant no VH-NG-523) for providing financial support to his research group. We gratefully acknowledge partial financial support from the German Research Foundation (DFG) through SFB677 (C01) and SFB 877 (Project B2, and Cluster of Excellence 'Inflammation at Interfaces', EXC306-PMTP1 and EXC306-PWTP2) to M.E. and D.A., respectively.

## Author contributions

M.E. conceived the ideas, designed and performed the basic experiments, and led the project. R.A. performed most of the materials synthesis and characterizations under M.E. supervision. D.D.-Z., S.H., and M.A. contributed to materials synthesis and character-izations. J.S., D.A., and M.E. designed the experiments for the biological part and D.D.-Z. performed the biological experiments under their supervision. L.K. and T.D. performed TEM characterizations. M.E. wrote the paper draft that has been shared, commented, and edited by all co-authors.

## Additional information

**Competing interests:** The authors declare no competing financial interests.

**Publisher's note**: 

