## [Peer Review File · Nature Communications]

Reviewers' comments:

Reviewer #1 (Remarks to the Author):

This work presents a method to synthesize zinc peroxide nanoparticles (NP) via a process route inspired by underwater chemistry, and an investigation of their cytotoxicity. Novelty in this work as presented is claimed in three areas: a new synthetic route to NP, the concept of the NP "swimming", and the results concerning cytotoxicity.

As presented, it is not at all clear that the repeated use of the term "swimming" to describe the behavior of the clusters produced is appropriate, or in agreement with the currently accepted terminology used within colloidal science. Consulting with any one of a number of review and experimental articles in the area of colloidal motion, will reveal that "swimming" behavior refers to colloids that are generating an additional motion producing thrust. This term is consequently applied to a range of systems, including actuated devices that have moving parts, and autonomous, or field actuated devices powered by mechanisms such as self-diffusiophoresis, self-electrophoresis, thermophoresis or bubble production.

The reported observations and discussion does not provide evidence for this being the case within the colloidal system they describe, and so calling these colloids "nanoswimmers" is not justified. For this claim to be substantiated, the authors must provide experimental evidence or suggest a plausible mechanism that fits within current literature to explain how the Zn NP are able to generate thrust within their fluid environment. It appears to be more likely that the motion observed by the authors is not a self-generated "swimming" effect, but instead a feature of the motion, or flow of the fluid the colloids are immersed in. Indeed, due to the temperature gradients, convective flow is very likely. It is established convention that such motion is not described as swimming; in short: not all moving colloids are swimming! However, if re-phrased using the correct terminology (including the title being modified), this criticism could be addressed and still leave a paper with sufficient novel content to be of interest.

Based on this possibility, I also offer some additional suggestions

Phrases such as "the greenest solvent ever" P1 Line 33, do not seem appropriate for a technical publication.

Also "green" is overused without explanation,

e.g.

P2. Line 2 "tune their size for green functions" - what is a green function? an example is needed

Also the conjecture that

P2. Line 10 "they must be synthesized via a green approach" is hyperbole, it is clear that there are other non-green routes to make NP which could be applied. This statement should be softened to reflect the fact that a green approach is desirable but not imperative.

Also, the inclusion in the results section and first Figure of some simple experiments to illustrate the Leidenfrost phenomena is unusual, normally Figures would be reserved for novel findings, and a previously reported phenomena would be introduced by reference to other literature.

In addition, Line 23-27 Page 2 stick out as a statement of track-record by the authors that does not appear to align with information the reader needs to understand the paper.

Figure 1: a+b Poor quality images, dark and hard to see, required more labels. Figure 1c, again needs more labels, also the "nanoswimmers" are drawn with a two tone Janus structure, which many genuine Janus swimmers have, but which the authors do not mention or discuss this feature in their generated NP, this could lead to significant confusion together with the terminology they adopt.

Figure 2 a is also very unclear, much better images are required to realistically communicate the colloidal motion effect the authors describe

Reviewer #2 (Remarks to the Author):

This study addresses the green and scalable synthesis route of nanoparticles (nps) with the aid of a newly introduced concept: underwater Leidenfrost dynamic chemistry. The overall reaction scheme to form nps takes place in three steps: (i) initial chemical reaction and formation of nps within the underwater two-phase overheated zone, (ii) formation of nanoclusters, (iii) swimming of nanoclusters towards the colder regions away from the heated surface for further growth. As claimed by the authors, this dynamic chemistry enables the production of nps with a very narrow size distribution, and eliminates the need for additives or surfactants. As one example, the authors have tested the synthesis of zinc peroxide nps using this route, and tested their cytotoxicity effect against cancer cells. As shown by the graphs, the cytotoxicity of nps against different cells depends on the size distribution of the synthesized np, highlighting the importance of size-tailored synthesis route suggested earlier. Overall, the project is very interesting and the manuscript is well written. However, there is still much room for improvement.

1) The choice of zinc peroxide to be synthesized by the green underwater Leidenfrost is very persuasive, as it has therapeutic properties. Also, the concept of underwater Leidenfrost introduced earlier in the manuscript can be extended to other chemical synthesis routes involving water as the solvent. However, in order to be a highly promising route for nanomedicine, the authors need to clarify that the synthesis of zinc peroxide shown here is not an isolated example.

2) Unlike the method introduced here, it seems that the synthesis of nps using droplet Leidenfrost may offer less control over the size distribution of nps, and probably this is an important selling point of this study. However, I couldn't see this highlighted in the manuscript. The authors need to clearly compare and contrast their current method with their previous work known as droplet Leidenfrost chemistry (refs. 12 for instance). This is essential as it improves the manuscript in that the readers won't feel it is only a scaled-up synthesis of their previously published results.

3) Page 3, line 3: hydraulic jump is a wave phenomenon that happens on a liquid surface. When the speed of the liquid equals the speed of the wave (and in opposite directions), the wave is stationary forming a "hydraulic jump". Therefore, the use of hydraulic jump to explain the droplet shape seems inappropriate here. Please elaborate or provide references.

4) Page 3, line 5: "Leidenfrost" is usually used for droplet boiling and if it happens in pool boiling, it is usually called "film boiling". It might be ok for the authors to keep calling it "underwater Leidenfrost phenomenon", but they should clarify this point. Also Figure 1c should clearly show a vapor film instead of "nanoclusters".

5) Fig. 2b, the best way to show underwater Leidenfrost phenomenon is to image a stable vapor film covering the surface. The rise of carbon fibres does not proof the existence of the Leidenfrost film. The fibres could be simply raised by vapor bubbles.

6) The quality of some images is not good at all: Fig.1 (all of them), Fig. 2a (barely could see what is it in there). I also did not find any videos.

7) Page 5, line 3-14: This should come when the authors discuss the demonstration of underwater

Leidenfrost concept on page 2 starting from line 20.

8) Equation (1) has two identical parameters with various definitions which are needed to be fixed, and clarified.

9) Authors please comment in more details how to avoid the formation of zinc oxide in the process of peroxide formation.

10) What is the possible range of temperature for the overall process to occur? Can the synthesis of nps take place in different regimes of boiling (nucleate or film boiling) with more or less the same outcome? These need to be addressed in the manuscript.

Reviewer #3 (Remarks to the Author):

The authors present their findings on the preparation of spherical ZnO₂ nanoparticles with dimensions from >50 nm to >450 nm. The preparation of these particles utilizes Leidenfrost chemistry to prepare these materials from zinc acetate in aqueous solutions. A proposal for the growth of these particles is proposed, which is distinct from a La Mer growth of nanoparticles. The particles were analyzed by a XRD and electron microscopy, and their biocompatibility and cell toxicity analyzed for a series of cancer cell lines. The work is interesting, and the results intriguing, but the work is not ready for publication.

Their process produces a series of different sized spherical particles. The authors claim the preparation of ZnO₂ nanoparticles based on the analysis of their products using X-ray diffraction analyses, but these results could overlook the presence of amorphous ZnO or concentrations of ZnO that are below the detection limits of XRD. It does not appear that other analytical techniques were used to verify the stoichiometry of Zn and O in the final products and to verify the composition of the particles. Additional, complementary techniques, include X-ray photoelectron spectroscopy, inductively coupled plasma mass spectrometry.

The Leidenfrost technique is intriguing for its ability to prepare spherical particles of sizes that are proportional to the concentration of reagents in solution, but the impacts of variations in the duration of heating are not apparent from the studies presented by the authors. Do the particles grow in size or change their apparent uniformity of size and shape with prolonged heating? What are the durations of the heating processes that were presented in these studies? Many of the experimental details are missing from the report in its current form (e.g., concentrations of reagents used, purity of the water, and duration of the synthetic processes). Further details are needed for others to reproduce this work.

The characterization of the spherical products includes electron microscopy and particle size analysis (using light scattering techniques). The particle size analysis by either technique needs to be performed in more detail. The current particles were analyzed, for the most part, using only a few particles to indicate the diameter of a population of particles. It is important that proper population distributions be demonstrated for each of the different sizes of particles, as this is a key contribution to the results of the cell studies according to the authors. Further light scattering data would greatly strengthen these studies, and further microscopy data is needed on finely dispersed particles with histograms associated with the measured particle dimensions.

A detailed size distribution analysis is needed in the case of all samples as further indicated in the results in Figure 3. Therein we can observe a bimodal distribution of particles. Are all the samples bimodal in their particle size distribution? Further statistical analyses are needed to help indicate the products and processes taking place.

In Figure 3a, the authors provide a plot of changes in charge presumably as a function of reaction time. The addition of reagents is presumably around the spike in the curve associated with the Leidenfrost condition, but the details are lacking to sufficiently interpret the plots within this figure. The axes require labels and the plots need further explanation within the text, including a brief overview of these measurements (even though the details are referenced to other literature, without context it is challenging at best for the readers to interpret this data without context).

The authors present fast Fourier transforms from their high resolution TEM data, and use this as evidence of the composition of the nanoparticles assembled into hollow spherical structures. These structures are interesting and the mechanism of their formation are even more interesting. The analysis requires further support as the rings indicated in the FFTs are unclear (even with magnifying the images in the manuscript these are too difficult to discern). Further FFT analysis and support from further HRTEM data is warranted. They should also discuss further details, such as the calibration standards used for referencing their sample analysis.

The authors note the presence of ZnO in the hollow spherical sample of nanoparticles and attribute its presence (over that of ZnO₂) to sample degradation under the electron beam conditions. It is standard practice in the field of electron microscopy analyses to perform a series of dose studies and to identify the optimal conditions for imaging and analysis of beam sensitive samples, as well as to prove that the operator is avoiding conditions that lead to sample damage. It is unclear if the authors have performed these studies prior to the TEM analysis of their materials.

The cell toxicity studies indicate a general trend of increasing cell death with increased mass concentration of particles in suspension. There is sometimes an opposite trend observed between the two analyzed sizes (426 and 126 nm) for their impact on triggered cell death. The results are interesting with regards to this differentiation. It is unclear how these trends compare with similar concentration dependent dose studies for ZnO_x species with similar cell lines. A number of studies can easily be found in the literature from a quick literature search. Their results should be compared and contrasted in detail to the prior results in the field. Further details are required for proper interpretation of the error bars in these plots and the statistical analyses used when analyzing the results. Other details include the number of passages of each type of cell line studied, the levels of dissolved Zn²⁺ in solution, levels of surfactant in solution, and methods used to introduce the particle to the cell cultures. No control samples were provided as a comparison to other materials with more well-known cell response factors.

The purity of the ZnO₂ nanoparticles was not properly proven. A variety of complementary analytical techniques are required to verify the results. As suggested above, free ions in solution are of interest, as are the concentrations of ligands remaining in solution from the acetate, and the potential presence of ZnO as observed by TEM analysis. Complementary techniques include the use of XPS, ion chromatography techniques, ICP-MS, etc.

In general, the manuscript lacks an in-depth discussion. The actual discussion section is one paragraph and most of the manuscript is devoted to presenting the data. A detailed account of the interpretation of the results is necessary, putting the results of these studies into the context of the field, such as alternative synthetic routes (e.g., comparing with other preparations for ZnO₂; such as RSC ADVANCES, 2016, 6 (88), 84777-84786, the authors of which have studied their products in detail that include conditions for release of oxygen with hydrolysis near physiological conditions or thermal decomposition). A similar context is necessary for the cell toxicity studies.

Further details are required in the experimental methods, which currently do not include sufficient details for others to reproduce the work. The particle size analysis should include further details, such

as the types of solutions used, concentrations of particles in solution, temperature(s) of the solution, theoretical model used to interpret the data, and type of sample holder used.

Another series of details needed include how the particles were "released" from the reaction substrates for TEM analysis. Was this by sonication? Or agitation? The operating conditions for the TEM analysis are also missing, as are the XRD operating conditions (e.g., source, accelerating potential, XRD sample holder, and software used in the analysis and background correction techniques). Further experimental details are also needed for the other types of analyses.

The minor concerns include standardization of formatting of the references, and avoiding the inclusion of "extreme" terminology, such as "greenest solvent ever"... "fabrication of pure... nanoparticles". The authors suggest that their materials can be utilized in "selective killing of cancerous tissues", but this has not been proven. They also suggest that their preparation is "free from surfactants and additives", which were not true as the solution contains zinc acetate leaving acetate in solution and possibly free zinc ions following the synthesis. They also claim a "narrow size distribution", but this needs further data to support this statement as outlined above. A series of grammatical issues also need to be corrected.

Reviewer #1:

Reviewer #1- Comment 1: As presented, it is not at all clear that the repeated use of the term "swimming" to describe the behavior of the clusters produced is appropriate, or in agreement with the currently accepted terminology used within colloidal science. Consulting with any one of a number of review and experimental articles in the area of colloidal motion, will reveal that "swimming" behavior refers to colloids that are generating an additional motion producing thrust. This term is consequently applied to a range of systems, including actuated devices that have moving parts, and autonomous, or field actuated devices powered by mechanisms such as self-diffusiophoresis, self-electrophoresis, thermophoresis or bubble production. The reported observations and discussion does not provide evidence for this being the case within the colloidal system they describe, and so calling these colloids "nanoswimmers" is not justified. For this claim to be substantiated, the authors must provide experimental evidence or suggest a plausible mechanism that fits within current literature to explain how the Zn NP are able to generate thrust within their fluid environment. It appears to be more likely that the motion observed by the authors is not a self-generated "swimming" effect, but instead a feature of the motion, or flow of the fluid the colloids are immersed in. Indeed, due to the temperature gradients, convective flow is very likely. It is established convention that such motion is not described as swimming; in short: not all moving colloids are swimming! However, if re-phrased using the correct terminology (including the title being modified), this criticism could be addressed and still leave a paper with sufficient novel content to be of interest.

Authors: We appreciate the informative and corrective comment of the reviewer. Agreeing with his/her comment concerning inactive nature of the particles, we omitted the term of "nanoswimmers" in the title and throughout the manuscript. But, since the paper

deals with a specific “swimming” behaviour of colloids (please see Supplementary Videos 3&4) based on the Leidenfrost state and wave patterned vortex forming at the hot/cold fluids interface (please see IR camera images; Figure 3b), rather than convection, we kept the term of “swimming”. Noteworthy, as we found out through a literature survey, this term is not solely about active colloids and can be used to describe the behaviour of inactive ones as well (*Scientific Reports* 5, Article number: 8546 (2015), doi:10.1038/srep08546).

In our study, hydrodynamic nature of the Leidenfrost phenomenon ensures swimming of the nanoclusters towards the much colder region giving rise to growth of monodisperse particles. This trend can be controlled and utilized for creation of size tailored nanoparticles. The uniqueness of this process was pointed out (please see Supplementary Videos 1&2) while showing how a flattened liquid on a colder substrate is converted to a droplet or loose carbon fibres fly together as a coherent black layer upward in the cold region. Accordingly, we’d rather discuss such a term within the manuscript based on our findings to clarify the origin of swimming of the clusters as following:

Page 6, lines 16-29: *“the confined dynamic chemistry underneath the water is self-regulated by a thermocapillary driven hydrodynamic response, as proven by snapshots of the IR camera. In particular, Figure 3b shows development of the confined state and temperature gradient within a Petri dish that has been suddenly introduced to a hot plate at 300°C. It is obvious that a temperature gradient in analogous to that seen in the Leidenfrost droplet [12] has been established. The highest temperature exceeding 100°C exists in the overheated zone of the two-phase region (liquid/steam). Also, induced by the temperature gradient, a wavy interface between the lower (hot) and upper (less hot or “colder”) part of the fluid is created, which is responsible for the hydrodynamic response and hence swimming of the clusters (as visualized in Figure 2a and Supplementary video 3). In this context, the gradient environments act as a generalized force to separate the nucleation and macroscopic growth of the clusters. Under this circumstance, the overheated zone provides an optimal condition for a confined reaction wherein the high evaporation rate leads to local supersaturation hence nucleation and primary growth of the clusters. Induced by the hydrodynamic response, the primary formed clusters, acting*

as seeds, will then swim to the colder region wherein further interaction and growth happen”.

Page 3, lines 12-18: *”It is obvious that the hydrothermal vents essential to release the pressure at the interface form and the carbon fibres are splashed towards the colder region. The emergence and growth of the hydrodynamic vents are governed by the Rayleigh–Taylor instability owing to different densities of the overheated water (low density) and the cold water (high density) above. Following the generation of the overheated zone, a wave patterned vortex appears at the hot/cold fluids interface mainly due to the Kelvin–Helmholtz instability and thereby the black layer flows upward in the cold region”.*

Reviewer #1- Comment 2: Phrases such as "the greenest solvent ever" P1 Line 33, do not seem appropriate for a technical publication. Also "green" is overused without explanation, e.g. P2. Line 2 "tune their size for green functions" - what is a green function? an example is needed. Also the conjecture that P2. Line 10 "they must be synthesized via a green approach" is hyperbole, it is clear that there are other non-green routes to make NP which could be applied. This statement should be softened to reflect the fact that a green approach is desirable but not imperative.

Authors: Respecting to the reviewer’s comment, such an adjective was replaced in some cases and softened in others as follows:

Page 1 line 35: *“the greenest solvent ever”* was replaced with *“an absolutely eco-friendly solvent”*.

Page 2 lines 1 and 19: *“green”* was replaced with *“eco-friendly”*.

Page 2 line 13: *“green”* was replaced with *“sustainable”*.

Page 2 lines 3-5: The statement was modified as: *”The particular relevant challenge is to design a method able not only to green synthesize nanoparticles [3-7] but also to tune their size for highly demanded functions in biomedicine, in cancer treatment, for instance”.*

Page 2 lines 12-14: The statement was modified as: “*To employ peroxide nanoparticles in biomedical and therapeutic applications, it is preferred to synthesize them via a sustainable approach in a pure form i.e. free from organic molecules, surfactants and additives and in a narrow size distribution.*”

Reviewer #1- Comment 3: Also, the inclusion in the results section and first Figure of some simple experiments to illustrate the Leidenfrost phenomena is unusual, normally Figures would be reserved for novel findings, and a previously reported phenomena would be introduced by reference to other literature.

Authors: In response to this comment, we need to mention that the results represented as Figure 1a are about a new set of Leidenfrost experiments and completely novel. In the Leidenfrost scheme, as a step further towards scalability of the idea and extending its applicability to any substrate and any medium, here as shown in Supplementary Video 1, we prove for the first time that even a *flattened solution droplet (film) mounted on a cold Si substrate can undergo the process*. This point was stressed in the Figure 1a caption, and declared in the manuscript as follows:

Page 2, line 29-Page 3, line 7: “*The Leidenfrost phenomenon encompasses several forms and is recognized by a minimum in the boiling curve, called as the Leidenfrost point ¹⁴. With respect to each form, the phenomenon can be called differently. For instance, the Leidenfrost state of water droplet is known as the film boiling state in a pool boiling process. Thus, this reaction is not necessarily performed in a droplet, while as patented by us the Leidenfrost chemistry can be carried out in other media e.g., in an aqueous solution ¹⁵. In fact, the dynamic Leidenfrost phenomenon depends on the heat flux supplied by the interface and the thickness of the thermal boundary layer of the fluid. Figure 1a and supplementary video 1 visualize this effect where **a small cold water film on a colder substrate is suddenly introduced to a superheated plate at 300°C and the liquid is suddenly converted to a typical Leidenfrost droplet**. Upon spending the incubation time, the adherent layer of the liquid becomes superheated and hydrothermally explodes owing to the sudden conversion to steam. Meanwhile, primary vapour bubbles are formed and start to grow. Vaporization spreads out forward and the adherent liquid*

on the superheated substrate is expanded to vapour, causing a sudden drop in the hydrostatic pressure and the liquid is re-shaped as a droplet. The overheated state of the Leidenfrost phenomenon occurs and the droplet is now levitated.”

Reviewer #1- Comment 4: In addition, Line 23-27 Page 2 stick out as a statement of track-record by the authors that does not appear to align with information the reader needs to understand the paper.

Authors: Respecting to the corrective comment of the reviewer, such statement was removed from the manuscript, but its former sentence was completed as:

Page 2, lines 26-28:” *Recently, our research activities have notably contributed in resolving of the Leidenfrost phenomenon in terms of specification of the involved mechanism based on electrostatic and thermocapillary processes [12]’.*

Reviewer #1- Comment 5: Figure 1: a+b Poor quality images, dark and hard to see, required more labels. Figure 1c, again needs more labels, also the "nanoswimmers" are drawn with a two tone Janus structure, which many genuine Janus swimmers have, but which the authors do not mention or discuss this feature in their generated NP, this could lead to significant confusion together with the terminology they adopt.

Authors: Respecting to the Reviewer’s opinion, we significantly enhanced the quality of the images. Also, we included some more informative labels including temperature of the hot plate whereon the flattened solution droplet (film)/Si substrate and beaker containing carbon nanofibers are mounted. Moreover, the figure caption was modified as follows below.

Regarding the sketch (Figure 1c), the shape of the nanoparticles was somehow modified to avoid any confusion with Janus particles. The term of “*nanoswimmers*” was also replaced with “*monodisperse peroxide nanoparticles*”. In addition, to clarify the mechanism, a vapour film was depicted at the bottom of the beaker.

Figure 1. The underwater Leidenfrost phenomenon. (a) and (b) the snapshots (1 cm scale bar) illustrate formation of a levitated droplet (70 μL) when a flattened cold droplet (film) mounted on a colder Si substrate is exposed to a superheated plate at 300 $^{\circ}\text{C}$ (the upper row), and flight of carbon fibres (0.03 g) induced by the Leidenfrost effect under water (5 mL) and emergence of a vapour film flowing upward (the lower row), (c) the sketch shows the process of formation and clustering of nanoparticles from the overheated zone towards the colder region, respectively.

Reviewer #1- Comment 6: Figure 2 a is also very unclear, much better images are required to realistically communicate the colloidal motion effect the authors describe.

Authors: Respecting to the Reviewer's idea, we enhanced the quality of the images. Moreover, for the sake of representing important images such as Figure 2a, in a higher scale, we excluded Figure 2f (SEM image of the synthesized monodisperse ZnO_2 nanoparticles by the beaker approach at low magnification) in the previous version and transferred it to the supplementary information part as Supplementary Figure 1. We

believe in the current format, the image obviously demonstrates formation and clustering of the particles on the surface of water bath.

Reviewer #2:

Reviewer #2- Comment 1: The choice of zinc peroxide to be synthesized by the green underwater Leidenfrost is very persuasive, as it has therapeutic properties. Also, the concept of underwater Leidenfrost introduced earlier in the manuscript can be extended to other chemical synthesis routes involving water as the solvent. However, in order to be a highly promising route for nanomedicine, the authors need to clarify that the synthesis of zinc peroxide shown here is not an isolated example.

Authors: We appreciate the useful suggestion of the reviewer. As he/she mentions, in this study, the aim is synthesis and application of zinc peroxide nanoparticles for cancer therapy. But, in a broader spectrum, our green nanofabrication method, has shown a

promising applicability in synthesis of a diverse range of functional nanomaterials including metal, metal oxide as well as peroxide particles that referring to them can be out of the scope of the paper. In this study, we not only synthesize zinc peroxide nanoparticles through this novel approach but also for the first time, demonstrate its applicability as cancer nanotherapeutics. According to the reviewer's suggestion, we added the following statement to the manuscript:

Page 9 lines 27-29: "*Noteworthy, in a broader spectrum, our green nanofabrication method, has also shown a promising applicability in synthesis of a diverse range of other functional nanomaterials including metal, and metal oxide ones*".

Reviewer #2- Comment 2: Unlike the method introduced here, it seems that the synthesis of nps using droplet Leidenfrost may offer less control over the size distribution of nps, and probably this is an important selling point of this study. However, I couldn't see this highlighted in the manuscript. The authors need to clearly compare and contrast their current method with their previous work known as droplet Leidenfrost chemistry (refs. 12 for instance). This is essential as it improves the manuscript in that the readers won't feel it is only a scaled-up synthesis of their previously published results.

Authors: We appreciate the useful suggestion of the reviewer. As he/she points out, in the classic or primary version of the Leidenfrost process, extreme rotation of the solution droplet and its complicated dynamics suppresses control over size distribution of the synthesized particles. In contrary, here, we were able to adjust the process, leading to formation of monodisperse nanoparticles. Respecting to the reviewer's suggestion and to highlight this part, we added SEM of nanoparticles formed in the droplet as supplementary Figure 4 while adding the following statement to the manuscript:

Page 5 lines 14-17: "*The high monodispersity of the synthesized nanoparticles is undoubtedly a large advantage for the presented version of the Leidenfrost process over the classic one relying on a superfast spinning and wandering droplet with complicated*

involved dynamics hampering creation of monodisperse particles (Supplementary Figure 4)”.

Supplementary Figure 4. SEM images of heterodisperse ZnO₂ nanoparticles synthesized by the classic Leidenfrost technique at two different magnifications. The nanoparticles were synthesized based on a precursor solution of 10 mM zinc acetate.

Reviewer #2- Comment 3: Page 3, line 3: hydraulic jump is a wave phenomenon that happens on a liquid surface. When the speed of the liquid equals the speed of the wave (and in opposite directions), the wave is stationary forming a “hydraulic jump”. Therefore, the use of hydraulic jump to explain the droplet shape seems inappropriate here. Please elaborate or provide references.

Authors: We agree with the point raised by the reviewer. Accordingly, we removed this term and rephrased the statement as:

Page 3 lines 5-6: *“Vaporization spreads out forward and the adherent liquid on the superheated substrate is expanded to vapour, causing a sudden drop in the hydrostatic pressure and the liquid is re-shaped as a droplet”.*

Reviewer #2- Comment 4: Page 3, line 5: “Leidenfrost” is usually used for droplet boiling and if it happens in pool boiling, it is usually called “film boiling”. It might be ok for the authors to keep calling it “underwater Leidenfrost phenomenon”, but they should clarify this point. Also Figure 1c should clearly show a vapor film instead of “nanoclusters”.

Authors: Respecting to the reviewer's comment, we clarified this point in the manuscript (as following) and in the caption of Figure 1b.

Page 2, lines 30-32: "With respect to each form the phenomenon can be called differently. For instance, the Leidenfrost state of water droplet is known as the film boiling state in a pool boiling process".

In addition, the sketch (Figure 1c) is aimed to show the formation of the nanoclusters in the overheated zone that are breathed (erupted) to the colder region in a volcanic manner. For this reason, we kept it, but modified it to show the vapour film underneath the fluid, as the reviewer asked.

Figure 1. The underwater Leidenfrost phenomenon. (a) and (b) the snapshots (1 cm scale bar) illustrate formation of a levitated droplet (70 μ L) when a flattened cold droplet (film) mounted on a colder Si substrate is exposed to a superheated plate at 300 $^{\circ}$ C (the upper row), and flight of carbon fibres (0.03 g) induced by the Leidenfrost effect under water (5 mL) and emergence of a vapour film flowing upward (the lower row), (c) the sketch shows the process of formation and clustering of nanoparticles from the overheated zone towards the colder region, respectively.

Reviewer #2- Comment 5: Fig. 2b, the best way to show underwater Leidenfrost phenomenon is to image a stable vapor film covering the surface. The rise of carbon fibres does not proof the existence of the Leidenfrost film. The fibres could be simply raised by vapor bubbles.

Authors: Respecting to the reviewer's comment, we depicted such a vapour film covering the surface (please refer to the above comment). In response to the second part of this comment, it is worth to mention that the carbon fibers are not firmly integrated as a

coherent film rather are as loose fibres. If the bubbles were responsible for raise of the fibers, as seen in Figure 1b (the part related to 20 sec), individual fibers could behave differently. But, this is not the case and after 68 sec, the whole fiber mat flies implying presence and interference of a vapour film flowing upward where the Leidenfrost state occurs. Such a point was already discussed thoroughly in the manuscript as:

Page 3 lines 2-7: *“Upon spending the incubation time, the adherent layer of the liquid becomes superheated and hydrothermally explodes owing to the sudden conversion to steam. Meanwhile, primary vapour bubbles are formed and start to grow. Vaporization spreads out forward and the adherent liquid on the superheated substrate is expanded to vapour, causing a sudden drop in the hydrostatic pressure and the liquid is re-shaped as a droplet. The overheated state of the Leidenfrost phenomenon occurs and the droplet is now levitated.”.*

Page 3 lines 16-20: *“Following the generation of the overheated zone, a wave patterned vortex appears at the hot/cold fluids interface mainly due to the Kelvin–Helmholtz instability and thereby the black layer flows upward in the cold region. These videos and images (supplementary video 2 and Figure 1b, respectively) illustrate the hydrodynamic nature of the Leidenfrost phenomenon and the vapour generation underneath the water in an arbitrary form”.*

Reviewer #2- Comment 6: The quality of some images is not good at all: Fig.1 (all of them), Fig. 2a (barely could see what is it in there). I also did not find any videos.

Authors: Respecting to the reviewer’s idea, we significantly enhanced the quality of the images. In addition, the videos were uploaded again.

Reviewer #2- Comment 7: Page 5, line 3-14: This should come when the authors discuss the demonstration of underwater Leidenfrost concept on page 2 starting from line 20.

Authors: We immensely thank the reviewer’s suggestion, aiming to improve the quality of the paper. However, this part is about discussion of the results represented in Figure 3a, implying charge separation in the aqueous solution subject to a fast heating process

acting as the driving force of the Leidenfrost dynamic chemistry. Whereas, Page 2 line 20 onward is solely about general introduction of the Leidenfrost effect and possibility of generalizing it to further substrates and media. Thus, we strongly believe the mentioned part must be kept at its current place.

Reviewer #2- Comment 8: Equation (1) has two identical parameters with various definitions which are needed to be fixed, and clarified.

Authors: This equation has two parameters of v_+ , the minimum particle volume of the stable nuclei, and \dot{v}_+ that is the mean volume growth rate whose dot was hardly visible. Respecting to the reviewer's comment, we enhanced visibility of dot in the respective section.

Reviewer #2- Comment 9: Authors please comment in more details how to avoid the formation of zinc oxide in the process of peroxide formation.

Authors: The origin of formation of ZnO was solely improper high electron beam intensity applied during TEM measurements. Our new TEM measurements clearly prove that ZnO₂ is the only composition that is created through the process.

The first TEM investigations presented in the manuscript implying formation of ZnO along with ZnO₂ were based on high electron beam intensities (necessary for HRTEM data). As the above described products transform by electron beam irradiation, the pristine state was investigated using electron diffraction at low dose and short exposure time. To do this, JEOL 2010 was used with 200 kV LaB₆ cathode which has a significantly lower beam intensity compared to the FEG Tecnai F30. Time resolved series of electron diffraction (ED) patterns, obtained based on respective TEM images (Supplementary Figure 6), are shown in Figure 3f. As seen in the Figure, clearly sharp rings are found at the start of the measurements, including a reflection at $d = 1.756 \text{ \AA}$ being characteristic for ZnO₂ (see mark). However, after few seconds the sample reacts and the crystallinity decreases rapidly.

Supplementary Figure 6. Time resolved TEM images employed for production of the low dose illumination ED data.

Figure 3f. Time resolved ED data under low-dose illumination (~1 sec illumination time for each image; the marks are attributed to a reflection at $d = 1.756 \text{ \AA}$ being characteristic for ZnO_2).

To further prove this hypothesis and to confirm 1:2 stoichiometry of ZnO_2 , time-sequential EDX data were acquired. The EDX data acquisition sequences were: EDX measurement ~30 sec, ~60 sec waiting, and acquiring again an EDX spectra. The data for a representative measurement are given below.

Table 1. The results of the TEM-EDX measurements confirming 1:2 stoichiometry of ZnO₂.

Measurement	O atm%	Zn atm%
1	72.49	27.50
2	72.42	27.54
3	72.15	27.84

Based on the results obtained by XRD, TEM and TEM-EDX, the observed particles are ZnO₂ particles initially which quickly transform to hollow spheres made of ZnO and ZnO₂ nanoparticles upon electron beam irradiation.

We accordingly made changes to the manuscript:

Page 7, lines 22-Page 8, line 10:” *With respect to the chemical composition of ZnO₂, it is worth mentioning that, the products were unstable against the irradiating beam, quickly transforming from compact nanoparticles to hollow spheres of a ZnO/ZnO₂ mixture after irradiation. In Figure 3e, the final state of the particles is depicted while images of intermediate stages can be found in the Supplementary Information part (Supplementary Figure 6). HRTEM and FFT of a representative ca. 50 nm hollow nanoparticle further witnesses that the spheres consist of randomly oriented clusters. In absence of reflections being characteristic for one component most reflections can be indexed assuming ZnO and ZnO₂, respectively. However, d-values of 2.6027 Å and 1.9107 Å ((002)_{ZnO} and (012)_{ZnO}, respectively) are only consistent with ZnO structure, and d = 1.722 Å, 1.9886 Å, and 2.1784 Å ((022)_{ZnO₂}, (112)_{ZnO₂} and (102)_{ZnO₂}) are characteristic for ZnO₂. From the FFT, the strong peaks with d-spacing of 2.65 Å and 1.93 Å (marked b and f in Figure 3e bottom right) imply the presence of ZnO nanoparticles as the major component. Whereas, the weak peaks with d-spacing of 1.77 Å, 2.01 Å and 2.19 Å (marked yellow d, e and g in Figure 3e right, bottom) are attributed to the minor quantity of ZnO₂.*

As the above described products transform by electron beam irradiation, the pristine state was investigated using electron diffraction at low dose and short exposure time. Time resolved series of electron diffraction (ED) patterns are shown in Figure 3f. As

seen in the Figure, clearly sharp rings are found at the start of the measurements, including a reflection at $d = 1.756 \text{ \AA}$ being characteristic for ZnO_2 (see mark). However, after few seconds the sample reacts and the crystallinity decreases rapidly. Based on the results obtained by XRD and TEM, the observed particles are ZnO_2 particles initially which quickly transform to hollow spheres made of ZnO and ZnO_2 nanoparticles upon electron beam irradiation.”

Reviewer #2- Comment 10: What is the possible range of temperature for the overall process to occur? Can the synthesis of nps take place in different regimes of boiling (nucleate or film boiling) with more or less the same outcome? These need to be addressed in the manuscript.

Authors: We highly appreciate the reviewer for raising this important point. To address this question, we performed additional experiments to correlate temperature of the hot plate thus water bath to the induced charge in the bath reactor. Accordingly, we could register the temperatures relevant to the transition and onset of the Leidenfrost point at $150 \text{ }^\circ\text{C}$ and $250 \text{ }^\circ\text{C}$, respectively. The fast evaporation event at this stage is characterized by creation of the negatively charged fluid as shown in the following Figure. When temperature rises over $150 \text{ }^\circ\text{C}$, negative charges form in the medium and the salt precursor is gradually converted to the peroxide particles. As much as temperature increases i.e. different regimes of boiling, this conversion is expected to be higher, as shown in the SEM images. The images are attributed to the sample containing the major fraction of salt against particles (i.e. the transition state) and the sample containing the major fraction of particles against salt (onset of the Leidenfrost point), respectively.

Accordingly, we added such important findings to the manuscript as follows:

Page 5, lines 33-Page 6, line 9: *“To highlight the importance of this point while characterizing different boiling regimes through the charge concept, we performed additional experiments, whereby temperature of the hot plate thus water bath was correlated to the induced charge in the bath reactor (Supplementary Figure 5). Accordingly, we could register the temperatures relevant to the transition and onset of the Leidenfrost point at $150 \text{ }^\circ\text{C}$ and $250 \text{ }^\circ\text{C}$ respectively. The fast evaporation event at this*

stage is characterized by creation of the negatively charged fluid as shown in the Supplementary Figure 5. When temperature rises over 150 °C, negative charges form in the medium. As much as temperature increases i.e. different regimes of boiling, the magnitude of charges rises and thereby conversion of the salt precursor to the peroxide particles is promoted, as shown in the SEM images of Supplementary Figure 5. These results in fact highlight importance of the Leidenfrost state and its relevance to the nanochemistry.”

Supplementary Figure 5. Relationship between temperature of the hot plate thus water bath to the induced charge in the bath reactor (SEM images show the samples containing the major fraction of salt against particles and vice versa at the transition boiling state and at onset of the Leidenfrost state, respectively).

Reviewer #3:

Reviewer #3- Comment 1: Their process produces a series of different sized spherical particles. The authors claim the preparation of ZnO₂ nanoparticles based on the analysis of their products using X-ray diffraction analyses, but these results could overlook the presence of amorphous ZnO or concentrations of ZnO that are below the detection limits of XRD. It does not appear that other analytical techniques were used to verify the stoichiometry of Zn and O in the final products and to verify the composition of the particles. Additional, complementary techniques, include X-ray photoelectron spectroscopy, inductively coupled plasma mass spectrometry.

Authors: We thank the reviewer for his/her precious suggestion. To prove the stoichiometry of Zn and O in the synthesized nanoparticles, we performed TEM-EDX analysis. The data for a representative measurement are given below and imply formation of ZnO₂ nanoparticles.

Measurement	O atm%	Zn atm%
1	72.49	27.50
2	72.42	27.54
3	72.15	27.84

Reviewer #3- Comment 2: The Leidenfrost technique is intriguing for its ability to prepare spherical particles of sizes that are proportional to the concentration of reagents in solution, but the impacts of variations in the duration of heating are not apparent from the studies presented by the authors. Do the particles grow in size or change their apparent uniformity of size and shape with prolonged heating? What are the durations of the heating processes that were presented in these studies? Many of the experimental details are missing from the report in its current form (e.g.,

concentrations of reagents used, purity of the water, and duration of the synthetic processes). Further details are needed for others to reproduce this work.

Authors: In response to this comment, we should say the heating process was lasted for several minutes. Prolonging the heating, as we observed, could lead to Ostwald ripening of the particles and creation of heterodisperse particles, as seen in the Figure following this comment. Based on our record, the optimum heating duration giving rise to desirable morphology and monodispersity was 40 sec for a 10 mL solution while it can last for more than 2 min. in a 50 mL one.

Concerning the lack of experimental details, we refer the reviewer to Page 4, lines 20-30 wherein the conditions of experiments including heating duration, amount and concentration of reagents and temperature have all been mentioned. The only missing parameter was purity of water that was added to the revised version at:

Page 5, lines 7-8: *“Noteworthy, the water used in all experiments as the main solvent was double deionized water (conductivity of 0.055 $\mu\text{S}/\text{cm}$)”*.

Moreover, the name of manufacturers of all the chemicals used in our experiments were added to the manuscript.

Reviewer #3- Comment 3: The characterization of the spherical products includes electron microscopy and particle size analysis

(using light scattering techniques). The particle size analysis by either technique needs to be performed in more detail. The current particles were analyzed, for the most part, using only a few particles to indicate the diameter of a population of particles. It is important that proper population distributions be demonstrated for each of the different sizes of particles, as this is a key contribution to the results of the cell studies according to the authors. Further light scattering data would greatly strengthen these studies, and further microscopy data is needed on finely dispersed particles with histograms associated with the measured particle dimensions. A detailed size distribution analysis is needed in the case of all samples as further indicated in the results in Figure 3. Therein we can observe a bimodal distribution of particles. Are all the samples bimodal in their particle size distribution? Further statistical analyses are needed to help indicate the products and processes taking place.

Authors: Respecting to the reviewer's comment, we added the details of particle size analysis and measurement of average particle size through SEM. Moreover, particle size distribution graphs were obtained and included in the supplementary information part. The histograms associated with the measured particle size based on the microscopy images were also included in the supplementary information part.

Page 10, line 28-Page 11, line 4: *"The size distribution of the ZnO₂ nanoparticles was determined using a ZetaSizer Particle Size Analyzer (Nano-ZS equipped with a red laser (633 nm, 4 mW) and a detection angle of 173 °, Malvern Instrument Ltd., Malvern, England) and an avalanche photodiode detector. The size of the nanoparticles in water suspensions at ambient temperature was measured based on the Dynamic Light Scattering (DLS) method. The size distribution was quantified as the relative volume of particles in size bands presented as size distribution curves (Malvern DTS 5.10 software). The suspensions were the Leidenfrost aqueous suspensions (2 mL; 1:1 diluted with de-ionized water) made after 40 sec exposure of the precursor solution (5, 10, 20 and 70 mM inversely proportional to the particle size) to a 300 °C hot plate. For each sample*

suspension, DLS measurements were made with a fixed run time of 20 sec. The scattering angle was set at 90°.

Moreover, the histograms of average particle size were determined from 100 random particles in an arbitrarily chosen area in the SEM images (Figures 2 f,g,h,b and i)''."

Supplementary Figure 2. Particle size distributions of ZnO₂ nanoparticles with average size of a) 70 nm (poly dispersity index (PDI)=0.095); b)126 nm (PDI=0.03); c) 220 nm (PDI=0.005); d) 680 nm (PDI=0.12)(Low PDI implies monodispersity of the nanoparticles).

Supplementary Figure 3. Histograms of particle size distributions of ZnO₂ nanoparticles with average size of a) 70 nm; b) 126 nm; c) 220 nm; d) 426 nm; e) 680 nm (extracted from SEM images labeled as Figure 2g,f,h,b and i, respectively).

Regarding bimodal particle size observed in Figure 3d, it must be noted that the sample taken for imaging was in a specific state wherein majority of particles are in a small regime. While, only a minor part belongs to coarse grown particles. Thus, in general, particles are monodisperse. Here, the idea was to follow up the growth of the particles over time. To avoid confusion, the statement concerning bimodal spherical zinc peroxide particles was removed and a more accomplished explanation was replaced as following: Page 7, lines 18-22: *“This hypothesis was confirmed by fishing the swimming nanoclusters of the 10 mM experiment on a silicon substrate and TEM grid at different time intervals. SEM image (Figure 3d) of the clusters reveals presence of tiny and large particles formed at onset of the experiment (after 10 sec). The time dependence growth is further confirmed by TEM observation of ultrafine nucleated zinc peroxide clusters, formed during the 10 mM experiment (after 1 sec)”*.

Reviewer #3- Comment 4: In Figure 3a, the authors provide a plot

of changes in charge presumably as a function of reaction time. The addition of reagents is presumably around the spike in the curve associated with the Leidenfrost condition, but the details are lacking to sufficiently interpret the plots within this figure. The axes require labels and the plots need further explanation within the text, including a brief overview of these measurements (even though the details are referenced to other literature, without context it is challenging at best for the readers to interpret this data without context).

Authors: We appreciate the reviewer for his/her constructive comment and apologize for the incomplete graph representing variation of charge in the reactor versus time. Accordingly, we corrected and discussed the graph (Figure 3a) in the manuscript. The details of the measurement was also added, as follows:

Page 5, lines 20-26: *“Figure 3a demonstrates charge separation in the aqueous solution (the 10 mM experiment; 50 mL) subject to a fast heating process. As seen in this figure, negatively charged water is created as similarly observed in our previous experiment based on the Leidenfrost droplet¹². The details of the charge measurements have been described elsewhere¹², but in general, a metal tungsten tip was connected to a grounded electrometer and then placed in an aqueous solution. The tip was located at a distance of 1 mm from the bottom of the aqueous solution containing Petri dish exposed to the high temperature of 300 °C and charges were registered”.*

Reviewer #3- Comment 5: The authors present fast Fourier transforms from their high resolution TEM data, and use this as evidence of the composition of the nanoparticles assembled into hollow spherical structures. These structures are interesting and the mechanism of their formation are even more interesting. The analysis requires further support as the rings indicated in the FFTs are unclear (even with magnifying the images in the manuscript these are too difficult to discern). Further FFT analysis and support from further HRTEM data is warranted. They should also discuss further details, such as the calibration standards used for referencing their sample analysis.

Authors: Respecting to the invaluable suggestion of the reviewer, we enhanced the quality of the relevant image and included further information related to the technique. The microscope used was fully calibrated by gold standard samples for HRTEM and ED data. EDX data analysis was calibrated by k-factors which are specific for the detectors used. The used EDX detector was also an EDAX system with Si/Li detector. Due to additional necessary investigations, we employed JEOL JEM-2100 whose details were also added to the manuscript while obtained results were presented in the supplementary information part.

Accordingly, we modified the relevant sections in the manuscript as follows:

Page 7, lines 31-Page 8, line 2: *“From the FFT, the strong peaks with d-spacing of 2.65 Å and 1.93 Å (marked b and f in figure 3e bottom right) imply the presence of ZnO nanoparticles as the major component. Whereas, the weak peaks with d-spacing of 1.77 Å, 2.01 Å and 2.19 Å (marked yellow d, e and g in figure 3e right, bottom) are attributed to the minor quantity of ZnO₂”.*

Page 11, lines 6-11: **TEM and SEM characterizations.** *TEM measurements were carried out using a FEI Tecnai F30 G2 STwin (FEG cathode, 300 kV, C_s=1.2 mm) equipped with a Si/Li EDX detector (EDAX-system). Further, a JEOL JEM-2100 (200 kV, LaB₆, C_s=1.0 mm) was also used for TEM observations at lower dose. Particles were transferred to a Cu lacey carbon TEM grid. SEM measurements were carried out using a*

Phillips XL 30 and a Field Emission SEM ZEISS ULTRA plus equipped with Oxford Instruments INCA-X-act EDS".

Reviewer #3- Comment 6: The authors note the presence of ZnO in the hollow spherical sample of nanoparticles and attribute its presence (over that of ZnO₂) to sample degradation under the electron beam conditions. It is standard practice in the field of electron microscopy analyses to perform a series of dose studies and to identify the optimal conditions for imaging and analysis of beam sensitive samples, as well as to prove that the operator is avoiding conditions that lead to sample damage. It is unclear if the authors have performed these studies prior to the TEM analysis of their materials.

Authors: We appreciate the precious comment of the reviewer that motivated us to probe this point. Respecting to his/her comment, we performed some additional experiments to investigate the effect of beam intensity on stability of the synthesized structures.

The first TEM investigations presented in the manuscript implying formation of ZnO along with ZnO₂ were based on high electron beam intensities (necessary for HRTEM data). As the above described products transform by electron beam irradiation, the pristine state was investigated using electron diffraction at low dose and short exposure

time. To do this, JEOL 2010 was used with 200 kV LaB₆ cathode which has a significantly lower beam intensity compared to the FEG Tecnai F30. Time resolved series of electron diffraction (ED) patterns, obtained based on respective TEM images (Supplementary Figure 6), are shown in Figure 3f. As seen in the Figure, clearly sharp rings are found at the start of the measurements, including a reflection at $d = 1.756 \text{ \AA}$ being characteristic for ZnO₂ (see mark). However, after few seconds the sample reacts and the crystallinity decreases rapidly.

Supplementary Figure 6. Time resolved TEM images employed for production of the low dose illumination ED data.

Figure 3f. Time resolved ED data under low-dose illumination (~1 sec illumination time for each image; the marks are attributed to a reflection at $d = 1.756 \text{ \AA}$ being characteristic for ZnO₂).

To further prove this hypothesis and to confirm 1:2 stoichiometry of ZnO₂, time-sequential EDX data were acquired. The EDX data acquisition sequences were: EDX measurement ~30 sec, ~60 sec waiting, and acquiring again an EDX spectra. The data for a representative measurement are given below.

Table 1. The results of the TEM-EDX measurements confirming 1:2 stoichiometry of ZnO₂.

Measurement	O atm%	Zn atm%
1	72.49	27.50
2	72.42	27.54
3	72.15	27.84

Based on the results obtained by XRD, TEM, and EDX-TEM the observed particles are ZnO₂ particles initially which quickly transform to hollow spheres made of ZnO and ZnO₂ nanoparticles upon electron beam irradiation.

We accordingly added the above mentioned results to supplementary information part, while referring and discussing them within the manuscript:

Page 7, lines 22-Page 8, line 10 :” *With respect to the chemical composition of ZnO₂, it is worth mentioning that, the products were unstable against the irradiating beam, quickly transforming from compact nanoparticles to hollow spheres of a ZnO/ZnO₂ mixture after irradiation. In Figure 3e, the final state of the particles is depicted while images of intermediate stages can be found in the Supplementary Information part (Supplementary Figure 6). HRTEM and FFT of a representative ca. 50 nm hollow nanoparticle further witnesses that the spheres consist of randomly oriented clusters. In absence of reflections being characteristic for one component most reflections can be indexed assuming ZnO and ZnO₂, respectively. However, d-values of 2.6027 Å and 1.9107 Å ((002)_{ZnO} and (012)_{ZnO}, respectively) are only consistent with ZnO structure, and d = 1.722 Å, 1.9886 Å, and 2.1784 Å ((022)_{ZnO2}, (112)_{ZnO2} and (102)_{ZnO2}) are characteristic for ZnO₂. From the FFT, the strong peaks with d-spacing of 2.65 Å and 1.93 Å (marked b and f in Figure 3e bottom right) imply the presence of ZnO nanoparticles as the major component. Whereas, the weak peaks with d-spacing of 1.77 Å, 2.01 Å and 2.19 Å*

(marked yellow d, e and g in Figure 3e right, bottom) are attributed to the minor quantity of ZnO₂.

As the above described products transform by electron beam irradiation, the pristine state was investigated using electron diffraction at low dose and short exposure time. Time resolved series of electron diffraction (ED) patterns are shown in Figure 3f. As seen in the Figure, clearly sharp rings are found at the start of the measurements, including a reflection at $d = 1.756 \text{ \AA}$ being characteristic for ZnO₂ (see mark). However, after few seconds the sample reacts and the crystallinity decreases rapidly. Based on the results obtained by XRD and TEM, the observed particles are ZnO₂ particles initially which quickly transform to hollow spheres made of ZnO and ZnO₂ nanoparticles upon electron beam irradiation.”

Reviewer #3- Comment 7:

Comment 7-1) The cell toxicity studies indicate a general trend of increasing cell death with increased mass concentration of particles in suspension. There is sometimes an opposite trend observed between the two analyzed sizes (426 and 126 nm) for their impact on triggered cell death. The results are interesting with regards to this differentiation. It is unclear how these trends compare with similar concentration dependent dose studies for ZnOx species with similar cell lines. A number of studies can easily be found in the literature from a quick literature search. Their results should be compared and contrasted in detail to the prior results in the field.

Authors: We very much appreciate that the reviewer considers our results interesting. As suggested, we have performed a literature search to uncover similar studies and to enable us to compare and contrast our data to these previous studies. Remarkably, we have been unable to locate a single previous publication in which ZnO₂ particles were used to induce death of cancer cells (search terms „ZnO₂“, „zinc dioxide“, „cancer“, „death“), underscoring the novelty of the data we present in our manuscript here.

As noted by the reviewer, there is a large number of studies using ZnO nanoparticles. In one of the most closely related studies we could find (Lin et al., J. Nanopart. Res. 11:25-39, 2009), the authors used human bronchoalveolar carcinoma-derived cells (A549) with 70 or 420 nm ZnO particles and they found that the particles of either size reduced the cell viability in a dose and time dependent manner (8-18ug/mL) by inducing oxidative stress without showing significant differences. In another related study (Kundu et al., J. Photochem. Photobiol. B. 140:194-204, 2014), the authors studied the effects of ZnO nanoparticles (100-120 nm average diameter) on HT29 colon carcinoma cells and normal peripheral blood mononuclear cells (PBMCs), concluding that that ZnO nanoparticles preferentially kill HT29 tumor cells over „normal“ PBMCs. This result is in contrast to our finding that for HT29 (as well as for Jurkat and Panc89) cells, the cytotoxic response induced by both sizes of ZnO₂ nanoparticles (426 and 126 nm) was comparable to that for PBMCs (Fig. 4 of our manuscript). However, the cytotoxic effects of ZnO₂ particles have not previously been compared to those of ZnO particles, and the differences may be a consequence of the different biological properties of the two kinds of nanoparticles. This is now discussed in more detail in the manuscript, and we again thank the reviewer for this suggestion.

Page 10, lines 10-23:” *In contrary to ZnO₂, there are numerous studies based on cytotoxic effect of ZnO, that could be regarded for the sake of comparison. In one of the most similar studies, Lin et al. [43] employed toxic effect of ZnO nanoparticles (70 and 420 nm) on human bronchoalveolar carcinoma-derived cells (A549). They found that with no significant difference the particles of either size can reduce the cell viability in a dose and time dependent manner (8-18 µg/mL) by inducing oxidative stress. In another related study, Kundu et al. [44] studied the cytotoxic effects of ZnO nanoparticles (100-120 nm) on HT29 colon carcinoma cells and PBMCs. They reported that ZnO nanoparticles preferentially kill HT29 tumor cells over „normal“ PBMCs. This result is in contrast to our findings as ZnO₂ nanoparticles, regardless of size, kill HT29 (as well as for Jurkat and Panc89) cells, as much as they kill PBMCs. Such a discrepancy is undoubtedly due to different interaction mode of ZnO₂ and ZnO nanoparticles with cells. Considering absence of any relevant study about biological performance of the ZnO₂ nanoparticles, we are unable to further discuss this point at the current moment. However, our research is*

ongoing and we are hoping to discover the probable involved mechanism in cytotoxicity effect of ZnO₂ particles to various cell types”.

Comment 7-2) Further details are required for proper interpretation of the error bars in these plots and the statistical analyses used when analyzing the results.

Authors: Again, we are grateful to the reviewer for pointing this out. We have performed each experiment independently a total of three times, where each point was measured a total of three times. In the manuscript, we show one representative experiment out of three. The error bars were calculated using the “STD” function of Microsoft Excel. Accordingly, we have added the following sentence to the legend of Figure 4: *“(Noteworthy in this Figure, one representative experiment out of three with three parallel determinations each is shown. In addition, error bars, calculated using the STD function of Microsoft Excel, indicate the corresponding standard deviations (SDs)).*

Comment 7-3) Other details include the number of passages of each type of cell line studied,

Authors: As outlined in the manuscript, HT29, Jurkat and U937 cell lines were originally purchased from ATCC, and L929Ts is a subline that was derived in the laboratory of Dieter Adam from the original L929 cell line (Ref. 45 of our manuscript). The cell line Panc89 was generated in the 1980s by Prof. Kalthoff (Ref. 46 of our manuscript) and obtained from him. Unfortunately, the exact number of passages for these cell lines has not been recorded. However, all cell lines were frozen in aliquots at early passages, and for each experiment, cells were rethawed from those early aliquots, used for experiments for a maximum of four weeks and then discarded. We have added a corresponding passage to the “Methods” section of our manuscript and we sincerely hope that this information is sufficient to answer the point raised by the reviewer.

Page 11, lines 24-28: *“PBMCs were isolated by Ficoll density gradient centrifugation from leukocyte concentrates. Leukocyte concentrates of healthy blood donors were obtained from the Institute for Transfusion Medicine of the University Hospital Schleswig-Holstein. All cell lines were frozen in aliquots at early passages, and for each experiment, cells*

were rethawed from those early aliquots, used for experiments for a maximum of four weeks and then discarded”.

In addition, further details concerning flow cytometry were added to the manuscript as follows:

Page 12, lines 14-23:” *The cells were re-suspended in PBS/5 mM EDTA containing 2 µg/ml of PI, and the red fluorescence was measured on a FACSCalibur flow cytometer (Becton Dickinson, Heidelberg, Germany). Each measurement was done in triplets with 10000 gated events. For Western blots, cells were harvested after treatment and lysed at 4 °C in TNE buffer (50 mM Tris pH 8.0, 1 % v/v NP40, 2 mM EDTA) supplemented with 10 µg/ml pepstatin/aprotinin/leupeptin, 1 mM sodium orthovanadate and 5 mM NaF. Identical amounts of protein per lane were resolved by electrophoresis on SDS polyacrylamide gels. After electrophoretic transfer to nitrocellulose, reactive proteins were detected using an antibody specific for PARP-1 (9542, Cell Signaling, Danvers, MA, USA), and the ECL detection kit (GE Healthcare, Munich, Germany). Equal loading as well as efficiency of transfer was verified by Ponceau S staining”.*

Comment 7-4) the levels of dissolved Zn²⁺ in solution, levels of surfactant in solution,

Authors: As mentioned earlier, no specific surfactant, reducing agent or any other chemicals were used in the whole process. In fact, this point is one of the main advantages of our techniques compared to other similar ones. In addition, for the sake of using the nanoparticles for the biological tests, the samples were three times washed then sedimented by centrifugation. Ultimately, dried ZnO₂ nanoparticles were re-suspended in sterilized water. Thus, the unreacted ions have all been removed before the final application.

Comment 7-5) and methods used to introduce the particle to the cell cultures.

Authors: Synthesized nanoparticles were washed three times with deionized water, and sedimented by centrifugation. Sedimented nanoparticles were dried at 40°C in ceramic crucibles in order to remove the excess water. Dried ZnO₂ nanoparticles were weighted

and re-suspended in sterilized water to give the desired stock solution concentration. At last, the nanoparticles were introduced into cell cultures in a total volume of 1mL. The stimulated cell cultures were incubated in a humidified incubator at 37°C and 5 % w/v CO₂ for 24 hours to conduct further characterisations. Such an explanation was added to the manuscript as:

Page 12, lines 4-9:"*The synthesized nanoparticles were washed three times with deionized water and sedimented by centrifugation. The sedimented nanoparticles were then dried at 40°C in ceramic crucibles to remove the excess water. The dried ZnO₂ nanoparticles were weighted and re-suspended in sterilized water to make the desired stock solution concentration. At last, the nanoparticles were introduced into cell cultures in a total volume of 1mL. The stimulated cell cultures were incubated in a humidified incubator at 37°C and 5 % w/v CO₂ for 24 hours to conduct further characterisations*".

Comment 7-6) No control samples were provided as a comparison to other materials with more well-known cell response factors.

Authors: We appreciate the reviewer for raising this issue. In this regard, control samples (blank samples, no nanoparticles) were included in these experiments that are labelled as "0" in Figure 4a-f. We now state this more explicitly in the figure legend, and we apologize for not making this sufficiently clear in the first place. With regard to comparing our data to other materials with more well-known cell response factors, please see our response to comment 7-1 above. To the best of our knowledge, the cytotoxic effects of ZnO₂ particles have not previously been compared to those of ZnO particles (or any other particles or materials), and any similarities or differences of the effects we observe with ZnO₂ particles to effects of other particles reported in the literature may result from the different biological properties of the different kinds of nanoparticles.

Reviewer #3- Comment 8: The purity of the ZnO₂ nanoparticles was not properly proven. A variety of complementary analytical techniques are required to verify the results. As suggested above, free ions in solution are of interest, as are the concentrations of ligands remaining in solution from the acetate, and the

potential presence of ZnO as observed by TEM analysis. Complementary techniques include the use of XPS, ion chromatography techniques, ICP-MS, etc.

Authors: In response to this comment, we would like to refer the reviewer to the answer to the comment 6 and comment (7-5). As proved through several supplementary characterizations, only pure ZnO₂ nanoparticles form by our technique. Appearance of ZnO phase was solely due to sensitivity of the particles to high beam intensity applied during TEM measurements. Furthermore the fabricated nanoparticles are several times washed as mentioned above (comment 7-5) and dried ZnO₂ nanoparticles have been used.

Reviewer #3- Comment 9:

Comment 9-1) In general, the manuscript lacks an in-depth discussion. The actual discussion section is one paragraph and most of the manuscript is devoted to presenting the data. A detailed account of the interpretation of the results is necessary, putting the results of these studies into the context of the field, such as alternative synthetic routes (e.g., comparing with other preparations for ZnO₂; such as RSC ADVANCES, 2016, 6 (88), 84777-84786, the authors of which have studied their products in detail that include conditions for release of oxygen with hydrolysis near physiological conditions or thermal decomposition).

Authors: We are not agree with the reviewer about generally poor discussion of the manuscript. As can be referred to synthesis and growth process of the nanoparticles (Pages 2-4), the Leidenfrost chemistry and various properties of the particles have been thoroughly discussed at different steps. Afterwards, yet, the discussion of biomedical function of the peroxide particles, due to novelty of this topic and lack of similar studies relying on peroxide nanoparticles, could be partly incomplete. In this regard, we are conducting new experiments whose results will be disseminated in the near future and justify the exact mechanism behind the killing behavior of the ZnO₂ nanoparticles versus ZnO ones.

Benefitting from the precious suggestion of the reviewer, we referred to the relevant literature and compared our results and advantages of our technique with similar ones as following:

Page 4, lines 3-19: "Conventional fabrication methods of ZnO₂ nanoparticles include hydrothermal synthesis, laser ablation, and sol-gel synthesis [17-19]. However, these methods involve chemicals such as different surfactants (e.g. SDS, CTAB, OGM and polyethylene glycol 200 (PEG 200)), thus could not be regarded purely green. In addition, as shown in TEM images, the generated particles are mostly aggregated, indicating a lack of nanoparticle stability [20]. To achieve ultrasmall and monodisperse ZnO₂ nanoparticles in aqueous phase, recently, a new synthesis route was reported by Bergs et al.[21]. In this approach, zinc acetate dihydrate was oxidized with hydrogen peroxide in aqueous media using high-pressure impinging-jet reactor. The high process pressure of 1400 bar and a specially-formed reaction chamber gave rise to short reaction times enabling fast nucleation and limited growth of nanoparticles, thus formation of ultrasmall ZnO₂ nanoparticles. Despite the advantages of this technique in terms of formation of monodisperse nanoparticles, use of complicated and energy consuming instruments and chemicals such as the stabilizing agent of bis[2-(methacryloyloxy)ethyl]phosphate is challenging. Advantageous over such methods, here, we demonstrate a completely eco-friendly fabrication route for peroxide nanoparticles without involvement of chemicals e.g. organic molecules, surfactants and stabilizers. More importantly, the entire process is accomplished in a small reactor i.e. a water bath exposed to a hot plate, implying simplicity and cost/energy efficiency of the process".

Comment 9-2) A similar context is necessary for the cell toxicity studies.

Authors: Even though the origin of cytotoxicity caused by nanoparticles is not completely understood, generation of reactive oxygen species (ROS) is thought to be the cause. Size and shape of the nanoparticles used would strongly influence their toxic potential. In general, as the particle size decreases, more ROS generation is expected due to increased surface defects of nanoparticles, decreased nano crystal quality, and higher

electron donor-acceptor impurities. Cytotoxicity is not only linked to nanoparticle characteristics but also depends on the cell type. Also, for our research we observed different cytotoxic profiles for suspension and adherent cells for different sizes of ZnO₂ nanoparticles. This work is unique in terms of using ZnO₂ nanoparticles as cancer therapeutics. For that reason we are unable to compare similar results however we can compare our results with regards to ZnO nanoparticles. Such a discussion was added to the manuscript:

Page 9, line 34-Page 10, line 23:” *Regarding the former dependency, even though the origin of cytotoxicity caused by nanoparticles is not completely understood, generation of reactive oxygen species (ROS) is thought to be the cause. Size and shape of the nanoparticles used would strongly influence their toxic potential. In general, as the particle size decreases, more ROS generation is expected due to increased surface defects of nanoparticles, decreased nanocrystal quality, and higher electron donor-acceptor impurities. However, as aforementioned, cytotoxicity is not only linked to nanoparticle characteristics but also depends on the cell type. This fact was the case in our research and we observed different cytotoxic profiles for suspension and adherent cells for different sizes of ZnO₂ nanoparticles. Considering uniqueness of our work in terms of the type of the nanoparticles being used as cancer therapeutics i.e. peroxides, it is difficult to clarify this effect with no previous knowledge concerning similar systems.*

In contrary to ZnO₂, there are numerous studies based on cytotoxic effect of ZnO, that could be regarded for the sake of comparison. In one of the most similar studies, Lin et al. ⁴³ employed toxic effect of ZnO nanoparticles (70 and 420 nm) on human bronchoalveolar carcinoma-derived cells (A549). They found that with no significant difference the particles of either size can reduce the cell viability in a dose and time dependent manner (8-18 µg/mL) by inducing oxidative stress. In another related study, Kundu et al. ⁴⁴ studied the cytotoxic effects of ZnO nanoparticles (100-120 nm) on HT29 colon carcinoma cells and PBMCs. They reported that ZnO nanoparticles preferentially kill HT29 tumor cells over „normal“ PBMCs. This result is in contrast to our findings as ZnO₂ nanoparticles, regardless of size, kill HT29 (as well as for Jurkat and Panc89) cells, as much as they kill PBMCs. Such a discrepancy is undoubtedly due to different interaction mode of ZnO₂ and ZnO nanoparticles with cells. Considering absence of any

relevant study about biological performance of the ZnO₂ nanoparticles, we are unable to further discuss this point at the current moment. However, our research is ongoing and we are hoping to discover the probable involved mechanism in cytotoxicity effect of ZnO₂ particles to various cell types”.

Reviewer #3- Comment 10: Further details are required in the experimental methods, which currently do not include sufficient details for others to reproduce the work. The particle size analysis should include further details, such as the types of solutions used, concentrations of particles in solution, temperature(s) of the solution, theoretical model used to interpret the data, and type of sample holder used.

Authors: Respecting to the reviewer’s comment, as mentioned in response to comment 3, we added the details of particle size analysis and measurement of average particle size through SEM to the manuscript.

Reviewer #3- Comment 11: Another series of details needed include how the particles were “released” from the reaction substrates for TEM analysis. Was this by sonication? Or agitation? The operating conditions for the TEM analysis are also missing, as are the XRD operating conditions (e.g., source, accelerating potential, XRD sample holder, and software used in the analysis and background correction techniques). Further experimental details are also needed for the other types of analyses.

Authors: Respecting to the reviewer’s concern about lack of experimental details, we included a more comprehensive explanation about particle size analysis (please see answer to the comment 3), TEM and SEM (please see answer to the comment 5), as well as XRD as follow:

Page 11, lines 13-20:”*X-ray diffraction patterns (XRD) of the particles were obtained using a Seifert XRD 3000 PTS (RICH. SEIFERT & Co GmbH) with 2-circuit goniometer. All measurements were carried out with Cu- K α radiation ($\lambda=1.5418$ Å) operating at 40 kV*

and 30 mA at ambient temperature. To ascertain that monochromatic ray radiation is used in the measurements, a Ni filter or a monochromator was employed. The measurements were performed over an angular range of 25 - 80 ° to both planar thin films deposited on glass substrate. The software used in the analysis and background correction was Rayflex”.

Reviewer #3- Comment 12:

Comment 12-1)The minor concerns include standardization of formatting of the references, and avoiding the inclusion of “extreme” terminology, such as “greenest solvent ever”... “fabrication of pure... nanoparticles”.

Authors: We followed the format of the journal for referencing. In addition, to respect the reviewer’s comment regarding avoiding extreme words as mentioned by him/ her, such terms were replaced by other ones or softened as follow:

Page 1 line 35: “*the greenest solvent ever*” was replaced with “*an absolutely eco-friendly solvent*”.

Comment 12-2)The authors suggest that their materials can be utilized in “selective killing of cancerous tissues”, but this has not been proven.

Authors: For the sake of proving the selective killing ability, we refer the reviewer to the cytotoxic behavior of 426 nm particles against PBMC and L929T cells. As shown in Figure 4b Vs. 4f, at specific nanoparticle concentration of 50 µg/mL, while the particles have no adverse effect on normal cells (PBMC) (this can be verified when comparing their performance with control condition in the adjacent column) they clearly kill the cancerous cells of L929Ts.

Comment 12-3) They also suggest that their preparation is “free from surfactants and additives”, which were not true as the solution contains zinc acetate leaving acetate in solution and possibly free zinc ions following the synthesis.

Authors: The approach we suggest here is free of any capping or reducing agent conventionally used in synthesis of nanoparticles as proved already by several characterizations. The unreacted ions have been removed from the synthesized particles during washing, sonication and drying steps. This means that what finally remains is solely pure peroxide nanoparticles that can be used for biomedicine applications.

Comment 12-4) They also claim a “narrow size distribution”, but this needs further data to support this statement as outlined above.

Authors: The claim concerning narrow size distribution of the nanoparticles was further confirmed through additional DLS experiments, as mentioned thoroughly in comment 3. The relevant results were also presented in the supplementary information part.

Comment 12-5) A series of grammatical issues also need to be corrected.

Authors: We reviewed the article several times and edited the following grammatical errors:

Page 9, line 19: “Notably” was replaced with “Noteworthy”

Page 8, line 27: In the statement of "...depends mainly on particle size and also on the target cells' type", "also on" was removed.

Page 8, line 23: "as an indicator of cell death" was replaced with "as the indicator of cell death".

Page 4, line 25: "a temperature of" was replaced with "the temperature of"

Reviewers' comments:

Reviewer #1 (Remarks to the Author):

Despite the modifications, the use of the term swimming is still present in the title and used to describe phenomena observed. I remain uncertain that this is appropriate.

The reference provided by the authors (doi:10.1038/srep08546) to justify this use of the term does not provide the evidence to validate their use of this term. In Figure 3 of that paper clear evidence for a flow of fluid relative to the "swimming" droplet is provided. This distinguishes this paper from the work under consideration.

As far as I can discern the authors have no evidence for a flow of fluid relative to the colloid surfaces, instead it is simply the case that the surrounding fluid is moving, driven by a thermal mechanism, and this is advecting the colloids which are immersed in the solution. Language such as splashing, and hydrodynamics added by the authors support this interpretation that it is a fluid flow driven transport phenomena that is being observed. In this case, swimming is not appropriate as it would set a new precedent of use for this word, which is currently used only when an object generates a fluid flow relative to its body. As a simple macroscale analogy, a floating body carried by a sea current is not described as being able to swim.

Either clear evidence to show how the colloids are generating a fluid flow is required, or it would be much clearer to talk about advection by a thermally driven fluid flow.

Reviewer #2 (Remarks to the Author):

The revision is satisfactory.

Reviewer #3 (Remarks to the Author):

The authors have addressed most of my original concerns. I appreciate the time the authors have taken to address these concerns and for their detailed replies. A few aspects of these concerns still remain. Although the authors have argued that they have addressed these points, there are some important clarifications or inclusions that are still necessary as outlined below. These are all fairly minor, and should not take long to amend, but will greatly improve the clarity, reproducibility and impact of their work.

These are listed in no particular order:

1) The authors still retain the statement "pure" nanoparticles. The term pure is an overstatement. Purity is only as good as the measurements one uses to assess purity. The statement could be misleading to non-experts and should be removed from the manuscript.

2) They also insist that their particles are synthesized in a pure form free of organic molecules. Although they argue that the acetate ligands from their zinc are not serving as surfactants on the nanoparticles (which is unclear), they clearly have organic molecules in solution. This statement should be revised. The fact that they don't put any other additives in solution is clear, but the rest of this statement should be removed as it is misleading and in part untrue.

3) Furthermore, the authors claim that their 3x centrifugation process has adequately purified their final products of free ions (such as unreacted Zn ions or free acetate ions). There was not inclusion of adequate analytical techniques to prove this point. The statements regarding no free ions in solution should be removed or further supported. There are a number of techniques that one could use, which were originally suggested to the authors.

4) The authors indicate a purification of their nanoparticles by centrifugation, but the conditions (e.g., rpm and duration) from this process are missing.

5) For the "0" concentration of the blank or control, I appreciate the authors further clarification. Do these solutions contain the same supernatant as the nanoparticles? What is the composition of the solutions added as the control, excepting the inclusion of the nanoparticles?

6) Further details are necessary for the DLS analyses of the suspended nanoparticles. Possibly I missed it, but what was the cell holder / cuvette type used in these measurements? What specific model (within the Malvern software) was used to fit the data and what was the temperature of these measurements?

Reviewer #1:

Despite the modifications, the use of the term swimming is still present in the title and used to describe phenomena observed. I remain uncertain that this is appropriate.

The reference provided by the authors (doi:10.1038/srep08546) to justify this use of the term does not provide the evidence to validate their use of this term. In Figure 3 of that paper clear evidence for a flow of fluid relative to the "swimming" droplet is provided. This distinguishes this paper from the work under consideration.

As far as I can discern the authors have no evidence for a flow of fluid relative to the colloid surfaces, instead it is simply the case that the surrounding fluid is moving, driven by a thermal mechanism, and this is advecting the colloids which are immersed in the solution. Language such as splashing, and hydrodynamics added by the authors support this interpretation that it is a fluid flow driven transport phenomena that is being observed. In this case, swimming is not appropriate as it would set a new precedent of use for this word, which is currently used only when a object generates a fluid flow relative to its body. As a simple macroscale analogy, a floating body carried by a sea current is not described as being able to swim.

Either clear evidence to show how the colloids are generating a fluid flow is required, or it would be much clearer to talk about advection by a thermally driven fluid flow.

Authors: We appreciate the informative comment of the reviewer. To address this uncertainty, we need to point out some relevant facts:

- 1) The term of "swimming" was used in our manuscript solely based on our observation of the clusters' behavior in the water bath, swimming from the hot zone towards the cold one. This point can be seen clearly in supplementary video 4.

- 2) In our belief, “swimming” of the nanoclusters is a passive process, induced by the unique hydrodynamic behavior of the fluid undergoing the Leidenfrost process. During the process, thermal, pressure and concentration gradients caused by the Leidenfrost effect encourage the nanoclusters to swap and swim towards the colder region. Noteworthy, IR camera images shown in Figure 3b, witness the thermal gradient present in the water bath.
- 3) The “passive swimming” effect was very recently (right after submission of our revised version) verified by Jo et al. (*Physical Review E* **94**, 063116 (2016)). They state that flow oscillations can be used to passively actuate and control the motion of microswimmers. The almost unknown indirect actuation leverages the interactions between the fluid and body properties to create locomotion using partial or no direct control over the swimmer. More precisely, leveraging fluid-body interactions can lead to passive actuation of the swimmer via actuation of the fluid medium itself, such as via background flow oscillations with no direct control over the swimmer. Such actuation methods can play a vital role in future advanced industrial and medical applications, including minimally invasive surgery and cell sorting and manipulation. State of the art microrobots employ magnetic, electric, chemical, or optical forces for actuation. Similarly, in our approach thermal, pressure and concentration gradient present in the water bath subject to the Leidenfrost process could act as actuating forces for passive swimming of the clusters.
- 4) The hydrodynamic forces acting on swimmers are functions of the Reynolds number. As Joe et al. state swimming efficiency decreases monotonically with the amplitude and viscosity of the background flow. In our study, from the hot zone to cold one, viscosity suddenly increases. Thus at the cold zone, the nanoclusters are less mobile. While in the transition state between hot and cold zones, they are highly dynamic, due to large fluctuation of the background flow by the available gradients. This swimming process leads to shaping and size tuning of the nanoparticles.

Reviewer #3:

Reviewer #3-comment 1:

The authors still retain the statement "pure" nanoparticles. The term pure is an overstatement. Purity is only as good as the measurements one uses to assess purity. The statement could be misleading to non-experts and should be removed from the manuscript.

Authors: Respecting to the reviewer's comment, this term was removed from the entire manuscript.

Reviewer #3-comment 2:

They also insist that their particles are synthesized in a pure form free of organic molecules. Although they argue that the acetate ligands from their zinc are not serving as surfactants on the nanoparticles (which is unclear), they clearly have organic molecules in solution.

This statement should be revised. The fact that they don't put any other additives in solution is clear, but the rest of this statement should be removed as it is misleading and in part untrue.

Authors: Respecting to the reviewer's comment, we revised the mentioned statement as follows:

Page 2, lines 12-14: "*To employ peroxide nanoparticles in biomedical and therapeutic applications, it is preferred to synthesize them via a sustainable approach without involvement of additives and in a narrow size distribution*".

Reviewer #3-comment 3:

Furthermore, the authors claim that their 3x centrifugation process has adequately purified their final products of free

ions (such as unreacted Zn ions or free acetate ions). There was not inclusion of adequate analytical techniques to prove this point. The statements regarding no free ions in solution should be removed or further supported. There are a number of techniques that one could use, which were originally suggested to the authors.

Authors: We appreciate the useful suggestion of the reviewer. This claim was stated in response to the reviewer's comment in the former revision and has not been included in the manuscript.

Reviewer #3-comment 4:

The authors indicate a purification of their nanoparticles by centrifugation, but the conditions (e.g., rpm and duration) from this process are missing.

Authors: Respecting to the reviewer's comment, we added extra info concerning the centrifugation process, as follows:

Page 12, lines 10-11:" *Centrifugation of the nanoparticles was performed on a Hettich® EBA 20 centrifuge at 3461 g for 90 minutes. Subsequently, the nanoparticles were washed through sonication for 45 minutes*".

Reviewer #3-comment 5:

For the "0" concentration of the blank or control, I appreciate the authors further clarification. Do these solutions contain the same supernatant as the nanoparticles? What is the composition of the solutions added as the control, excepting the inclusion of the nanoparticles?

Authors: Control samples represent the cell cultures, whose details were previously mentioned in the methods part, in absence of nanoparticles. This means, in the case of PBMCs, Jurkat and U937, control samples contain only the respective cells cultured in PMI 1640 (Thermo Fisher Scientific, Dreieich, Germany) with 10% fetal bovine serum (FBS), 10 mM Hepes, and 10 IU/ml penicillin+streptomycin. With respect to the other

cell lines, controls include the respective cells cultured in DMEM (Thermo Fisher Scientific) with 5IU/mL Penicillin Streptomycin / L-glutamine (Thermo Fisher Scientific) and 10% FBS.

This point was clarified in the relevant Figure caption as follows:

Figure 4: “0` concentration represents blank (control) samples containing the cell culture media but in absence of ZnO₂ nanoparticles”.

Reviewer #3-comment 6:

Further details are necessary for the DLS analyses of the suspended nanoparticles. Possibly I missed it, but what was the cell holder / cuvette type used in these measurements? What specific model (within the Malvern software) was used to fit the data and what was the temperature of these measurements?

Authors: We appreciate the useful comment of the reviewer. In this regard, we added further details of the experiment to our manuscript as follows:

Page 10, line 28-Page 11, line 7:” **Particle size distribution measurements.** *The size distribution of the ZnO₂ nanoparticles was determined using a ZetaSizer Particle Size Analyzer (Nano-ZS equipped with a red laser (633 nm, 4 mW) and a detection angle of 173 °, Malvern Instrument Ltd., Malvern, England) and an avalanche photodiode detector. The size of the nanoparticles in water suspensions at ambient temperature was measured based on the Dynamic Light Scattering (DLS) method. The suspensions were the Leidenfrost aqueous suspensions (2 mL; 1:1 diluted with de-ionized water) made after 40 sec exposure of the precursor solution (5, 10, 20 and 70 mM inversely proportional to the particle size) to a 300 °C hot plate. For each sample suspension, DLS measurements were carried out in Plastibrand semi-micro poly(methyl methacrylate) cuvettes at 25 °C with a fixed run time of 20 sec. The scattering angle was set at 90°. The instrument recorded the intensity autocorrelation function, which was transformed into volume functions to obtain size information. The autocorrelation curves were fitted by the Malvern software (Malvern DTS 5.10 software). Two methods of analysis were used, including cumulants analysis to determine a mean size and polydispersity index and distribution analysis to determine actual size distribution. Using*

the fitted correlation functions, diffusion coefficients were obtained that were associated to hydrodynamic diameter via the Stokes–Einstein equation”.

Reviewers' comments:

Reviewer #1 (Remarks to the Author):

I remain unconvinced about the use of the term swimming in this manuscript, and think replacement by an alternative word would ensure there is no confusion about the origin of the motion phenomena reported. Replacing this term, which has a well defined meaning within the colloidal science community, with another (e.g. migration) would seem to have no negative impact on the overall presentation of the work, and remove this confusion. Significantly more work would be required to justify that there is a swimming effect present in the system under investigation.

I include comments on the rebuttal:

1) The term of "swimming" was used in our manuscript solely based on our observation of the clusters' behavior in the water bath, swimming from the hot zone towards the cold one. This point can be seen clearly in supplementary video 4.

I have no doubt that the clusters are moving as described in the manuscript, my only contention is that there are many fluid flow mechanisms that could produce this movement, and that the use of the term swimming is by convention reserved for a specific mechanism whereby the cluster itself generates a fluid flow (with, or without actuation), and the suggestion that this is present in the system under consideration is not supported by experimental evidence, or a plausible mechanism.

2) In our belief, "swimming" of the nanoclusters is a passive process, induced by the unique hydrodynamic behavior of the fluid undergoing the Leidenfrost process. During the process, thermal, pressure and concentration gradients caused by the Leidenfrost effect encourage the nanoclusters to swap and swim towards the colder region. Noteworthy, IR camera images shown in Figure 3b, witness the thermal gradient present in the water bath.

Again, I have no doubt that these gradients exist, but in order to justify the use of the term swimming, a link needs to be established between these gradients and a specific mechanism by which the clusters themselves can produce additional fluid flow. It still appears far more likely that it is the fluid itself that is moving due to these gradients, and the clusters are being carried along with the fluid. This is not usually referred to as swimming.

The "passive swimming" effect was very recently (right after submission of our revised version) verified by Jo et al. (Physical Review E 94, 063116 (2016)). They state that flow oscillations can be used to passively actuate and control the motion of microswimmers. The almost unknown indirect actuation leverages the interactions between the fluid and body properties to create locomotion using partial or no direct control over the swimmer. More precisely, leveraging fluidbody interactions can lead to passive actuation of the swimmer via actuation of the fluid medium itself, such as via background flow oscillations with no direct control over the swimmer. Such actuation methods can play a vital role in future advanced industrial and medical applications, including minimally invasive surgery and cell sorting and manipulation. State of the art microrobots employ magnetic, electric, chemical, or optical forces for actuation. Similarly, in our approach thermal, pressure and concentration gradient present in the water bath subject to the Leidenfrost process could act as actuating forces for passive

swimming of the clusters.

The use of this new reference to support the claim for swimming is surprising. The reference is a purely theoretical study that predicts swimming behavior for a very specific hypothetical v shaped device, able to deform in a particular prescribed way due to fluctuations in the surrounding fluid. As such there is no apparent link to a possible swimming effect in the authors work, where the clusters are clearly not likely to be arranged in this very specific geometric arrangement. The authors of the cited paper make no attempt to claim their observations can be generalized to a mechanism to produce swimming in any other geometrical arrangement of material. In fact, the two references the authors have used in the rebutals provide further evidence for the need to reserve the term "swimming" for systems that do possess this quite unusual and specific behavior.

The remainder of the comments simply restate that the swimming device field is an active research area with potential applications, which I agree with. However in the examples given there is a well established mechanism for how the actuation can produce phenomena that lead to fluid flow being generated by the colloid or device itself, and also evidence that the devices produce motion which cannot be explained solely by advection by a fluid flow. I still maintain that in the authors work sufficient evidence to make a plausible case for such a mechanism is absent.

4) The hydrodynamic forces acting on swimmers are functions of the Reynolds number. As Joe et al. state swimming efficiency decreases monotonically with the amplitude and viscosity of the background flow. In our study, from the hot zone to cold one, viscosity suddenly increases. Thus at the cold zone, the nanoclusters are less mobile. While in the transition state between hot and cold zones, they are highly dynamic, due to large fluctuation of the background flow by the available gradients. This swimming process leads to shaping and size tuning of the nanoparticles.

I have already compared the authors work to the reference, and cannot see any similarity due to the highly specific geometrical shape considered in the cited reference. The authors again mention flow in this reply, which makes me think that they also agree that fluid flow is likely to be responsible for the motion. This may be highly complex, and result in very interesting effects in the authors system, but I re-iterate that to refer to this as swimming without evidence or a clearly explained mechanism just adds unnecessary confusion to the manuscript that could be avoided by use of a more general term.

Reviewer:

I remain unconvinced about the use of the term swimming in this manuscript, and think replacement by an alternative word would ensure there is no confusion about the origin of the motion phenomena reported. Replacing this term, which has a well defined meaning within the colloidal science community, with another (e.g. migration) would seem to have no negative impact on the overall presentation of the work, and remove this confusion. Significantly, more work would be required to justify that there is a swimming effect present in the system under investigation.

The authors again mention flow in this reply, which makes me think that they also agree that fluid flow is likely to be responsible for the motion.

Authors:

We deeply appreciate the respectful reviewer for his/her diligence and the time spent on precisely discussing the disputed point. We have no doubt that the driving force of the phenomenon observed is the fluid's unusual behavior at the Leidenfrost temperature. We are absolutely aware that to make an object swim in a fluid, there should be asymmetrical streamlines of the fluid around the object. Here, the question is that how we can govern this condition. In our belief, it can be achieved by an active component of the system, i.e. the fluid and/or the object. The colloidal science community has ever dealt with a passive fluid and an active object. The object is asymmetrical thus stimulating accumulation of the fluid streamlines in one side more than the other. This effect gives rise to a pressure/velocity etc. difference, hence driving movement or swimming (i.e. decrease/increase pressure and increase/decrease velocity) of the object within the fluid. On the other hand, if there is a symmetrical object but the fluid is active, as seen in our study, a different situation applies. In particular, when the flow pattern is asymmetrical around a symmetrical object then a similar result is obtained and the object will swim passively.

The uniqueness of the Leidenfrost state originates from asymmetrical flow patterns of the fluid acting as the active component in the system. In fact, the object

is located within a transition region at interface of the overheated and cold zones wherein an asymmetrical flow pattern composed of a turbulent flow (overheated zone) and a laminar flow (colder region), as sketched below, form. The difference in pressure and velocity in the two regions (i.e. decrease/increase pressure and increase/decrease velocity in the overheated/cold zones, respectively) results in displacement of the cluster upwards by breaking the laminar flow lines. Otherwise, the clusters would remain at the bottom of the vessel.

Despite elaboration of our scientific point of view relying on the presented videos and IR camera images as evidences of our hypothesis, we came to this decision that at this stage we'd better change the term of "swimming" to "eruption". This flexibility facilitates spreading our observation and scientific achievements presented in our paper within a broader scientific community.